# REDO: Execution-Free Runtime Error Detection for COding Agents

## Abstract

As LLM-based agents exhibit exceptional capabilities in addressing complex problems, there is a growing focus on developing coding agents to tackle increasingly sophisticated tasks. Despite their promising performance, these coding agents often produce programs or modifications that contain runtime errors, which can cause code failures and are difficult for static analysis tools to detect. Enhancing the ability of coding agents to statically identify such errors could significantly improve their overall performance. In this work, we introduce *Execution-free Runtime Error Detection for COding Agents* (*REDO*), a method that integrates LLMs with static analysis tools to detect runtime errors for coding agents, without code execution. Additionally, we propose a benchmark task, *SWE-Bench-Error-Detection* (*SWEDE*), based on SWE-Bench (lite), to evaluate error detection in repository-level problems with complex external dependencies. Finally, through both quantitative and qualitative analyses across various error detection tasks, we demonstrate that REDO outperforms current state-of-the-art methods by achieving a 11.0% higher accuracy and 9.1% higher weighted F1 score; and provide insights into the advantages of incorporating LLMs for error detection.

## 1 Introduction

Large language models (LLMs) and LLM-based agents have exhibited significant potential in code generation, code editing, and code evaluation. This progress has culminated in the development of advanced LLM-based agents (hereafter referred to as *coding agents*) designed to address increasingly complex tasks. For example, SWE-Bench (Jimenez et al., 2024a) presents a demanding benchmark comprising repository-level coding challenges. This benchmark requires coding agents to generate a modification patch that solves a given problem within a GitHub repository, based on a problem statement expressed in natural language. To effectively navigate complex tasks such as those posed by SWE-Bench, coding agents must demonstrate proficiency in the following core competencies: 1) **comprehension** of the problem statement and retrieving relevant code, 2) **reasoning** towards a functionally correct solution, and 3) **generation** of programs free from runtime errors such as SyntaxError, AttributeError, or TypeError.

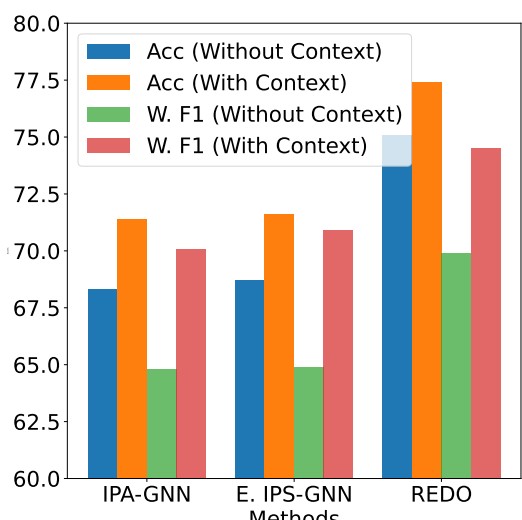

Figure 1: REDO outperforms current SOTA methods (Bieber et al., 2022) with respect to accuracy and weighted F1 (W.F1) on different tasks.

While the majority of coding agents across different tasks focus on enhancing comprehension, retrieval and reasoning capabilities, the systematic detection of runtime errors has received comparatively limited attention. However, ensuring that generated code is free from runtime errors is as criti-

cal as the aforementioned capabilities. For example, an AttributeError can cause the modified code to fail, irrespective of the agent's comprehension and reasoning processes. Indeed, coding agents are not immune to runtime errors. On SWE-Bench-lite (Jimenez et al., 2024b), the top six coding agents as of August 2024 (*CodeStory*, *Mentatbot*, *Marscode*, *Lingma*, *Droid*, *AutoCodeRover*) produce, on average, 1.8 SyntaxErrors, 25.8 TypeErrors, 5.2 NameErrors, and 11.3 AttributeErrors. Moreover, SWE-Bench (lite) is deliberately curated to include only straightforward problems, suggesting that the incidence of runtime errors could be substantially higher on the full SWE-Bench dataset. Additionally, detecting these runtime errors enables coding agents to rapidly iterate solutions, thereby reducing both time and cost.

Recent coding agents do often incorporate modules for detecting runtime errors. For example, *SWE-agent* (Yang et al., 2024) and *AutoCoderRover* (Zhang et al., 2024) employ static analysis tools such as Pylint (Foundation, 2024c), while *CodeR* (Chen et al., 2024) utilizes dynamic analysis by generating unit tests and executing the modified code. However, these methods have three critical limitations. First, because runtime errors are often triggered by specific inputs that are typically unpredictable from the code alone, static analysis approaches struggle to detect errors such as many TypeErrors. Second, although dynamic analysis techniques may identify these runtime errors, they require execution of the underlying code, which is problematic for three reasons. First, execution ceases upon encountering the first runtime error, allowing only one error to be detected at a time, thereby increasing the cost of detection. Secondly, dynamically configuring execution environments for diverse project setups and testing frameworks poses significant challenges and might trigger technical, legal or privacy concerns (Puddu et al., 2022; Lacoste & Lefebvre, 2023). Furthermore, although these mechanisms are commonly employed as separate modules, their performance has never been comprehensively evaluated. This lack of modular evaluation makes it difficult to assess the extent to which these error detection modules enhance the performance of coding agents.

In this work, we introduce an execution-free runtime error detection method, termed *REDO*, which integrates static analysis tools, such as Pyflakes (Foundation, 2024b) or PyRight (pyright, 2024), with a large language model (LLM). This approach extends the capabilities of static analysis tools to detect a broader range of errors without the need for code execution. The combination of these tools is specifically designed to maximize the advantages of both tools, achieving a balanced trade-off between reliability and the breadth of detectable errors. Similar to other works (Jimenez et al., 2024a; Bieber et al., 2022), we specifically focus on the runtime error detection in **Python** repositories; but our method could be straightforwardly extended to other languages. Moreover, we present a challenging and practical benchmark, *SWE-Bench-Error-Detection* (SWEDE), which is the **first** repository-level error detection in the presence of complex external dependencies. Beyond SWEDE, we perform a suite of experiments encompassing various tasks to further evaluate error detection algorithms. As shown in Figure 1, REDO significantly outperforms previous methods, obtaining SOTA performance across diverse scenarios; and through qualitative analysis, we provide insights into the underlying mechanisms of REDO.

## 2 RELATED WORK

### 2.1 LLM-BASED CODE GENERATION, CODING AGENTS, AND SWE-BENCH

LLMs (Ouyang et al., 2022; et al., 2024b; 2023b; 2024a; Anthropic, 2024) have been increasingly leveraged for automatic code generation (Nijkamp et al., 2023b; et al., 2021a; Chai et al., 2023; et al., 2024c; Nijkamp et al., 2023a; et al., 2023a; Gunasekar et al., 2023) and code repair (Xia et al., 2023; Shypula et al., 2024; Gunasekar et al., 2023; Prenner & Robbes, 2023; Huang et al., 2023). The advent of LLM-based agents has further expanded the scope of problem-solving in complex coding tasks, leading to the development of specialized coding agents. For example, *SWE-Agent* (Yang et al., 2024) is built on the *ReAct* framework (Yao et al., 2023) and incorporates a custom Agent-Computer Interface (ACI), enhancing the agent's ability to interact with the environment effectively. Similarly, *AutoCodeRover* (Zhang et al., 2024) provides an integrated set of search tools and includes *Pylint* within its toolkit, which serves to statically detect errors. Another significant contribution is *CodeR* (Chen et al., 2024), a multi-agent framework that facilitates code editing by coordinating multiple agents through structured graphs. Within this framework, specific agents like the *Reproducer* and *Verifier* are designed to generate unit tests and validate modified implementations, respectively. These coding agents are frequently evaluated using the *SWE-Bench*

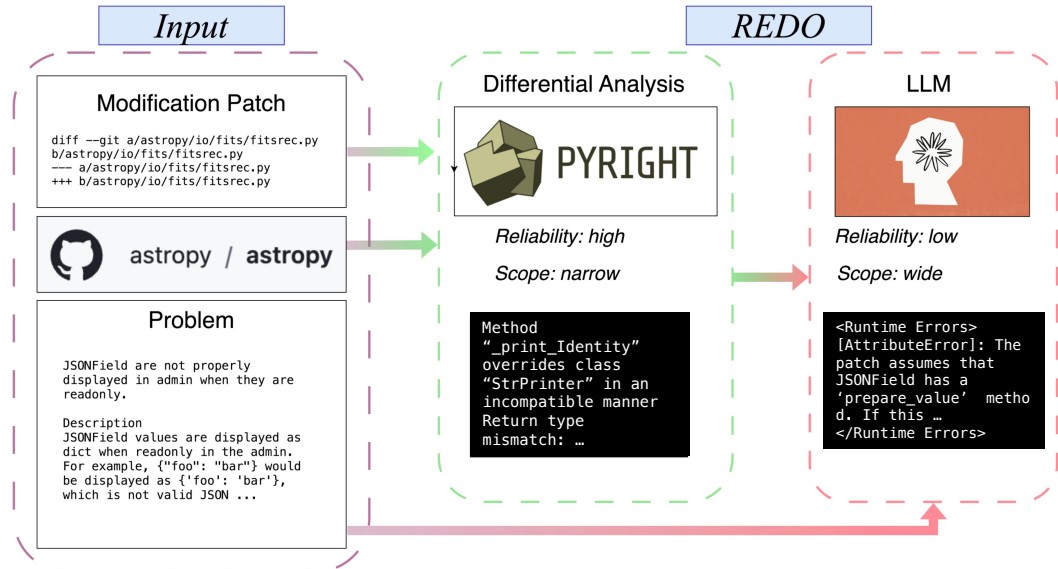

Figure 2: REDO employes a two-step process. When a repository-level modification is given, REDO first applies differential analysis to compare the original and modified implementations. If runtime errors are identified, REDO triggers an alert and rejects the patch. Conversely, if no errors are detected, the patch is forwarded to the LLM-based detection. The final determination regarding the patch's safety is made based on the detection provided by the LLM.

benchmark Jimenez et al. (2024a), which is composed of coding issues extracted from public GitHub repositories. This benchmark poses significant challenges by requiring coding agents to comprehend issues, localize relevant code segments, and produce correct modifications.

## 2.2 STATIC ANALYSIS TOOLS AND RUNTIME ERROR PREDICTION

Static analysis, a technique for examining computer programs without execution, is particularly valuable in contexts where executing the program might lead to legal, privacy, or computational concerns. Due to its non-executive nature, static analysis has found widespread application in error detection (Zheng et al., 2006; Dillig et al., 2007; Chow et al., 2024), bug identification (Ayewah et al., 2008; Mashhadi et al., 2024), and vulnerability discovery Charoenwet et al. (2024); Sonnekalb et al. (2023); Esposito et al. (2024); Chess & McGraw (2004); Livshits & Lam (2005); Evans & Larochelle (2002). In the context of Python programming, various professional static analysis tools have been developed to enhance code quality. For instance, *Pylint* (Foundation, 2024c) and *Pyflakes* (Foundation, 2024b) are designed to identify errors, while *Bandit* PyCQA (2024) focuses on detecting common security vulnerabilities. Additionally, tools such as *PyRight* (pyright, 2024) and *MyPy* (Foundation, 2024a) perform type checking, contributing to more robust software development. Although the integration of LLMs with static analysis is still in its infancy, some recent studies have proposed combining these technologies to enhance bug detection in complex systems, such as the Linux kernel (Li et al., 2024a), and to identify security vulnerabilities (Li et al., 2024b). However, the exploration of LLMs in conjunction with static analysis for runtime error detection remains limited. Additionally, Bieber et al. (2022) presents a notable effort in this domain by leveraging Graph Neural Networks (GNNs) for predicting runtime errors, along with proposing a dataset, referred to as *STA*, to evaluate their approach's efficacy.

## 3 EXECUTION-FREE RUNTIME ERROR DETECTION FOR CODING AGENTS

In this study, we introduce REDO, which serves to check the safety of modification patches. Here, "Unsafe" instances are those that might crash due to runtime errors; and "Safe" instances are those that can be successfully run. REDO operates through a two-phase process: differential analysis and LLM-based detection.

The differential analysis component employs a static analysis tool, which provides a dependable method for detecting runtime errors. However, its detection capabilities are generally constrained to

SyntaxError, AttributeError, and NameError. To address this limitation, the LLM-based detection is incorporated to reason about the input contexts. It extends REDO's detection capabilities to errors such as TypeError and ValueError, which are typically beyond the scope of static analysis.

By integrating these mechanisms, REDO achieves a balanced trade-off between reliability and the breadth of error detection. An overview of REDO's architecture is illustrated in Figure 2.

### 3.1 Differential analysis

Static analysis tools like Pyflakes and PyRight typically ensure detection through reliable methods, such as verifying syntax correctness and maintaining data type consistency, making them essential for identifying runtime errors in coding agents. In this study, we employ **PyRight** as our static analysis tool since it is fast and lightweight; however, our framework is designed to be flexible, allowing the integration of any static analysis tool.

Despite their utility, static analysis tools are affected by two challenges. First, they are prone to generating false positives, where potential vulnerabilities are incorrectly flagged (Kang et al., 2022; Kharkar et al., 2022; Murali et al., 2024). For example, when PyRight is applied to original python scripts containing the modified functions, which do not contain runtime errors, it falsely classifies an average of **267** instances, considering **89%** of all testing instances as "Unsafe" across various coding agents. To mitigate this issue, particularly in code edit tasks, we introduce the concept of *differential analysis*. This method involves applying static analysis tools to both the original and modified implementations separately. By comparing the errors detected in the original implementation ($S_{\mathrm{Orig}}$) with those in the modified implementation ($S_{\mathrm{Mod}}$), we can identify any new errors introduced by the modifications. If new errors are detected in the modified implementation, the patch is flagged as "Unsafe". Differential analysis effectively refines static analysis tools to focus specifically on runtime errors introduced by modifications, thereby filtering out false positives from the original implementation. Notably, this removes almost all positives induced by the original implementation.

The second challenge with static analysis tools is their inability to detect errors that are triggered under specific inputs. In dynamically typed languages like Python, variable data types are unknown before execution and also can change in response to different inputs. Since static analysis tools cannot reason about input contexts, they often fail to detect these errors, resulting in low recall in error detection performance. We show a concrete example in Figure 4. To address this challenge, we propose the LLM-based detection described in the next section.

### 3.2 LLM-based Detection

In comparison to static analysis tools, LLMs possess the ability to comprehend both the problem statement and the modification patch. This capability allows them to reason about potential input contexts and anticipate runtime errors that might be overlooked by static analysis tools. However, the reasoning process of LLMs is not always as reliable as the error detection mechanisms inherent in static analysis tools. To harness the complementary strengths of both approaches, we restrict the application of LLMs to instances deemed 'Safe' by static analysis tools. Accordingly, we have designed the LLM prompt template to identify potential runtime errors that static analysis tools may have missed.

Specifically, for each modification patch, we provide two additional pieces of information: the problem statement and the Python script containing the original version of the modified functions. The problem statement outlines the input contexts and describes the potential verification process for the modified implementation. The original implementation provides the running context of the modified functions, including the safe utilization of variables and functions. Subsequently, we prompt the LLM to enumerate potential runtime errors that may arise due to the modification and ultimately assess the safety of the patch. A detailed prompt template is provided in Appendix E.2.

Given the cost and latency associated with LLM calls, the LLM API is restricted to a single invocation per instance in our study. For the LLM, we utilize *Claude-SONNET-3-5*, with the temperature setting fixed at zero. A pseudo-code of whole REDO framework is given in Algorithm 1.

## 4 SWE-BENCH-ERROR-DETECTION (SWEDE)

The difficulty of runtime error detection can vary drastically among different coding problems. For instance, on the task proposed by Bieber et al. (2022), only one python script is considered on each data point. This task is practical as it resonates the competitive programming scenario where one python script should contain all functionality; and external dependencies are simple. However, as coding agents become more powerful, additional challenging and practical scenarios should be considered. In this work, we propose an repository-level error detection task, which is based on SWE-Bench (lite) (Jimenez et al., 2024a) and its evaluation results using different coding agents. We name this task as *SWE-Bench-Error-Detection* or *SWEDE*.

SWE-Bench (lite) is a popular benchmark dataset containing instances of repository-level coding problems. Taking a GitHub repository and a problem statement as inputs, SWE-Bench (lite) asks coding agents to generate a modification patch to resolve the problem. SWEDE extends SWE-Bench (lite) to include generated patches and evaluation logs from SWE-Bench leaderboard (Jimenez et al., 2024b); but with a focus on detecting runtime errors induced by generated patches, without executing the code.

The task is **challenging** for two reasons. First, the modified scripts usually call other python scripts, referred to as *external dependencies*, within the same repository. For instance, as shown in Table 1, when only one directory level above the location where the modified file resides is considered, there already are many dependencies on average. When all files in the repository are considered, the number of dependencies could become even more intimidating. Furthermore, the unit tests might not directly interact with the modified files. These two factors make SWEDE challenging for error detection algorithms as running contexts of variables and functions are harder to infer. The task is **practical** because, when coding agents autonomously modify repositories, detecting runtime errors early offers instrumental information; and can potentially reduce cost and time.

Table 1: Average External Dependencies in Parent Folder on SWEDE Across Coding Agents.

| Method | CodeStory | Demo | Aider | Lingma | Droid | ACR |
|---|---|---|---|---|---|---|
| Average dependencies | 5.56 | 4.56 | 6.07 | 5.75 | 5.61 | 5.98 |

As the problem is challenging, in this work, we focus on detecting if a modification patch will induce any runtime errors, e.g., SyntaxError, AttributeError, TypeError, etc. As a result, given a patch, we label it as **positive** if it contains an runtime error; and **negative** if the patch passes all unit tests or fails the unit tests only because of wrong functionality. We propose to measure the performance of error detection algorithms on SWEDE using precision, recall and F-1 score.

## 5 EXPERIMENTS

### 5.1 QUANTITATIVE RESULTS

**SWE-Bench-Error-Detection (SWEDE).** We first evaluate REDO on SWEDE task. We consider six State-of-the-art (SOTA) coding agents on the SWE-Bench-lite leaderboard, including *CodeStory*, *Mentatbot*, *Marscode*, *Lingma*, *Droid*, and *AutoCodeRover* (*ACR*) Zhang et al. (2024). We compare REDO to several baselines. First, we include two widely used static analysis tools, namely *Pyflakes* and *PyRight*. To eliminate false positive detection, we apply differential analysis (introduced in Section 3.1. Second, we include an LLM-only method, denoted as *LLM*. The LLM is prompted with the same template introduced in Section 3.2. Lastly, to study how the choice of static analysis tool affects REDO, we include a REDO framework with Pyflakes, named as *REDO-Pyflakes*. Due to the inherent stochasticity in generation, even with a temperature setting of zero, we generate the LLM responses three times and report the mean and standard deviation of the outcomes.

As shown in Table 2, static analysis tools (Pyflakes and PyRight) usually have higher precision scores, while LLM obtains better recall scores. This justifies our analysis on the difference between static analysis tools and LLM (as in Section 3). Furthermore, confusion matrices in Figure 3 show that the static analysis tool is more prudent in claiming unsafe patches, and therefore misses more failed patches than LLM does. By appropriately combining these two kinds of tools, REDO makes

a good trade-off between reliability and detectable score. This better trade-off contributes to superior F1 scores. Second, the performance of LLM is closer to REDO because REDO utilizes LLM on more than 70% instances. However, we note that despite the similarity, REDO makes less API calls because of the differential analysis step. We show confusion matrices using REDO and different coding agents in Figure 6. Also, as can be seen on the other dataset (STA) below, when static analysis tools performs better, REDO can significantly outperform the LLM-only baseline. Third, the performance of REDO-Pyflakes and REDO are close and outperform others, demonstrating the robustness of REDO against static analysis tool choices. We further report results using Anthropic Claude OPUS in Table 4 and SONNET-3.5 with temperature being 0.5 in Table 5. We can observe that REDO and REDO-Pyflakes consistently outperform static analysis tools and LLM, demonstrating the robustness of our proposed method against LLM configurations.

Table 2: Performance metrics by method and benchmark

| Coding agent | Metric | Method | | | | |
|---|---|---|---|---|---|---|
| | | Pyflakes | PyRight | LLM | REDO-Pyflakes | REDO |
| CodeStory | Precision | 32.1 | **43.7** | $34.0_{\ 0.2}$ | $33.3_{\ 0.2}$ | $33.9_{\ 0.2}$ |
| | Recall | 22.8 | 48.1 | $69.2_{\ 0.7}$ | $\underline{70.5}_{\ 0.7}$ | $\mathbf{75.5}_{\ 0.7}$ |
| | F1 | 26.7 | $\underline{45.8}$ | $45.6_{\ 0.4}$ | $45.3_{\ 0.3}$ | $\mathbf{46.8}_{\ 0.3}$ |
| Demo | Precision | **34.1** | $\underline{33.9}$ | $30.7_{\ 0.6}$ | $31.2_{\ 0.6}$ | $31.1_{\ 0.3}$ |
| | Recall | 34.9 | 45.3 | $61.2_{\ 1.8}$ | $\underline{70.2}_{\ 1.3}$ | $\mathbf{74.4}_{\ 1.2}$ |
| | F1 | 34.5 | 38.8 | $40.9_{\ 1.0}$ | $\underline{43.2}_{\ 0.8}$ | $\mathbf{43.8}_{\ 0.5}$ |
| Marscode | Precision | 22.9 | **33.8** | $28.7_{\ 1.0}$ | $27.9_{\ 0.7}$ | $\underline{29.1}_{\ 0.8}$ |
| | Recall | 16.4 | 37.3 | $68.7_{\ 1.5}$ | $\underline{74.6}_{\ 1.5}$ | $\mathbf{77.6}_{\ 1.5}$ |
| | F1 | 19.1 | 35.5 | $40.5_{\ 1.2}$ | $\underline{40.6}_{\ 1.0}$ | $\mathbf{42.3}_{\ 1.0}$ |
| Lingma | Precision | $\underline{45.8}$ | **48.3** | $40.5_{\ 0.1}$ | $40.3_{\ 0.2}$ | $41.3_{\ 0.2}$ |
| | Recall | 27.8 | 29.9 | $66.3_{\ 0.6}$ | $\mathbf{71.5}_{\ 0.6}$ | $\mathbf{71.5}_{\ 0.6}$ |
| | F1 | 34.6 | 36.9 | $50.3_{\ 0.2}$ | $\underline{51.5}_{\ 0.3}$ | $\mathbf{52.3}_{\ 0.3}$ |
| Droid | Precision | **53.3** | $\underline{42.1}$ | $41.8_{\ 0.3}$ | $40.1_{\ 0.3}$ | $42.0_{\ 0.3}$ |
| | Recall | 22.0 | 14.7 | $63.6_{\ 0.5}$ | $\underline{65.4}_{\ 0.5}$ | $\mathbf{68.2}_{\ 0.5}$ |
| | F1 | 31.2 | 21.8 | $\underline{50.4}_{\ 0.4}$ | $49.7_{\ 0.4}$ | $\mathbf{52.0}_{\ 0.4}$ |
| ACR | Precision | 45.3 | **54.7** | $\underline{39.2}_{\ 0.2}$ | $38.3_{\ 0.2}$ | $40.6_{\ 0.2}$ |
| | Recall | 23.5 | 34.3 | $65.7_{\ 1.0}$ | $\underline{69.3}_{\ 0.6}$ | $\mathbf{73.9}_{\ 0.6}$ |
| | F1 | 31.0 | 42.2 | $49.1_{\ 0.3}$ | $\underline{49.3}_{\ 0.0}$ | $\mathbf{52.4}_{\ 0.3}$ |

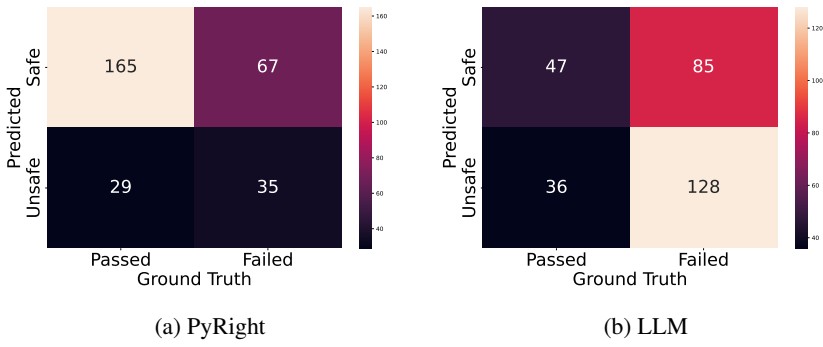

(a) PyRight (b) LLM

Figure 3: Confusion Matrices on CodeStory

**Ensemble patch** To demonstrate how the detection errors serve to enhance coding agents, we propose a preliminary algorithm to ensemble a *base* coding agent with an *auxiliary* agent, basing on the detection results from REDO. Specifically, given an instance in SWEDE, REDO first checks the safety of the patch $p_{\text{base}}$ from the base agent. If $p_{\text{base}}$ is safe, it will be accepted; otherwise, REDO will check the safety of the patch $p_{\text{aux}}$ from the auxiliary agent. If $p_{\text{aux}}$ is safe, it will be alternatively accepted; otherwise, the algorithm falls back to the base agent and accept $p_{\text{base}}$. Note that, since the top agents on the SWE-Bench leaderboard are proprietary, we are unable to run their methods to generate new patches. Instead, we utilize the patches reported in the SWE-Bench repository. A

pseudo-code of the ensemble algorithm is given in Algorithm 2. The quantitative results and analysis can be found in Section D.

**STA dataset.** To assess the general applicability of REDO, we conduct evaluations on another dataset: the balanced test set introduced by Bieber et al. (2022), hereafter referred to as *STA*. This dataset is derived from a code generation dataset CodeNet (et al., 2021b) and comprises approximately 27,000 Python submissions for competitive programming tasks. The balanced test set was designed to ensure that the number of submissions without errors is approximately equal to those containing runtime errors. The dataset categorizes 26 distinct error types, including the category "no error." The detailed enumeration of errors and their corresponding indices are consistent with those listed in the prompt template provided in Appendix E.3. Unlike the SWEDE dataset, each submission in STA consists of a single Python script that processes input via standard input (stdin) and performs its functionality without relying on external dependencies. In the context of STA, error detection algorithms are tasked with predicting runtime error types based on the Python submission, with or without the presence of input contexts. These input contexts describe the potential values that stdin may take, providing clues regarding possible runtime errors. We refer to the scenario involving input context as *With Context*, and the scenario without input context as *Without Context*.

Given that the task in STA differs from that in SWEDE, we adapt REDO using modified prompt templates. The templates for both the scenarios with and without input contexts are provided in Appendix E.3 and Appendix E.4, respectively. The performance of the algorithms is assessed using three metrics: accuracy (Acc), weighted F1 score (W.F1), and weighted error F1 score (E.F1). Weighted metrics are computed by calculating the F1 score for each class independently and averaged using a weight that depends on the number of true instances for each class; the E.F1 is calculated exclusively for data points containing a runtime error. We compare our methods with those reported by Bieber et al. (2022). To further investigate the contributions of individual components within REDO, we also evaluate *PyRight* and *LLM* separately. Since the original dataset samples do not include error-free submissions, we employed three different random seeds to sample an equivalent number of submissions for a fair comparison. The means are reported in Table 3, and the standard deviations are presented in Table 6.

First, as demonstrated in Table 3, REDO-PyRight achieves state-of-the-art (SOTA) performance across all six evaluated metrics, underscoring the superior performance of REDO in the context of STA. Additionally, we observed discrepancies between the annotated error types and the results obtained from our running results. Consequently, we also present evaluation results, enclosed in brackets, based on error types identified in our runs. Under these conditions, REDO exhibits even more pronounced improvements over the baselines. Second, our ablation PyRight obtains decent performance in both Without and With Context tasks. Considering that the input to STA submissions are usually most common cases, as opposed to corner cases in SWEDE, our result demonstrates that static analysis tools could perform well if the inputs and dependencies are simple. Furthermore, when comparing REDO to PyRight, we can also see that REDO obtains extra benefits by including the LLM based tool. This shows the effectiveness of our proposed framework.

**Moreover, the differing conclusions regarding static analysis tools and LLMs between SWEDE and STA highlight that each excels in distinct tasks.** By integrating these two tools, we can leverage the strengths of both, rendering our framework more versatile and effective across a broader range of tasks.

### 5.2 QUALITATIVE ANALYSIS

In this section, we qualitatively analyze how key design factors in REDO affects the performance. Consequently, we target answering the following two questions:

1. When does LLM succeed and fail to help REDO?
2. How could the detected errors help?

**When Does the LLM Succeed?** Figure 4 illustrates a case in which the LLM successfully identifies a runtime error that PyRight overlooked. This example is on the *django-13551* instance using coding agent *Lingma*. The complete LLM response and modification patch are presented in Figure 8. Specifically, the evaluation log in Figure 4 reveals that an *AttributeError* is triggered when the

Table 3: Quantitative results on detecting runtime errors in CodeNet dataset. Each method is evaluated with and without the input context. Three metrics are included, namely *accuracy* (Acc), *weighted F1 score* (W. F1), and weighted error F1 score (E. F1). **Bold values** indicate the best performance for each metric, while underlined values represent the second-best performance.

| Method | Without Context | | | With Context | | |
| --- | --- | --- | --- | --- | --- | --- |
| | Acc | W. F1 | E. F1 | Acc | W. F1 | E. F1 |
| GGNN | 62.8 | 58.9 | 45.8 | 68.3 | 66.5 | 56.8 |
| TRANSFORMER | 63.6 | 60.4 | 48.1 | 67.3 | 65.1 | 54.7 |
| LSTM | 66.1 | 61.4 | 48.4 | 68.1 | 66.8 | 58.3 |
| IPA-GNN | 68.3 | 64.8 | 53.8 | 71.4 | 70.1 | 62.2 |
| E. IPS-GNN | 68.7 | 64.9 | 53.3 | 71.6 | 70.9 | 63.5 |
| PyRight | 74.5 (76.4) | 67.2 (70.1) | 52.2 (54.2) | 74.5 (76.4) | 67.2 (70.1) | 52.2 (54.2) |
| LLM | 61.8 (64.1) | 54.0 (57.2) | 32.7 (35.8) | 64.3 (66.8) | 58.6 (62.1) | 40.4 (43.7) |
| REDO | **75.1** (76.8) | **69.9** (72.6) | **57.0** (59.1) | **77.4** (79.3) | **74.5** (77.5) | **64.3** (66.6) |

---

**LLM Prediction**

```
I'll analyze the modification patch for potential runtime errors:

<Runtime Errors>
[AttributeError]: The modification adds `user.email` to the hash value calculation. However,
not all user models in Django are guaranteed to have an email field. If a custom user model is
used that doesn't include an email field, this could raise an AttributeError.
</Runtime Errors>

<Conclusion> Unsafe </Conclusion>
```

**Evaluation log**

```
Traceback (most recent call last):
  File "/opt/pyenv/versions/3.6.15/lib/python3.6/unittest/case.py", line 59, in
testPartExecutor yield
  File "/opt/pyenv/versions/3.6.15/lib/python3.6/unittest/case.py", line 523, in subTest yield
  File "/opt/django__django/tests/auth_tests/test_tokens.py", line 57, in
test_token_with_different_email tk1 = p0.make_token(user)
  File "/opt/django__django/django/contrib/auth/tokens.py", line 28, in make_token return
self._make_token_with_timestamp(user, self._num_seconds(self._now()))
  File "/opt/django__django/django/contrib/auth/tokens.py", line 70, in
_make_token_with_timestamp self._make_hash_value(user, timestamp),
  File "/opt/django__django/django/contrib/auth/tokens.py", line 98, in _make_hash_value
return str(user.pk) + user.password + str(login_timestamp) + user.email + str(timestamp)

AttributeError: 'CustomEmailField' object has no attribute 'email'
```

Figure 4: Successful example

invalid attribute *email* is accessed on the variable *user*, which is an instance of a *CustomEmailField* object. PyRight failed to detect this runtime error due to the inability to infer the data type of *user* through static analysis. In contrast, the LLM, capable of reasoning about potential runtime contexts, successfully identifies the runtime error, thus marking the patch as *Unsafe*. This example demonstrates the advantage of leveraging LLMs to anticipate runtime errors. Another successful instance is documented in Appendix F.1.

**When Does the LLM Fail?** Figure 5 presents an instance where both PyRight and the LLM failed to detect a runtime error. This example pertains to the instance *django-11797* using coding agent *CodeStory*. The complete modification patch and the LLM's response are detailed in Figure 11. The runtime error occurs when an invalid column name is referenced in a query. Given the current input context, it is nearly impossible to infer the content of the query, leading to the LLM's failure to predict this runtime error. This example highlights the limitations of REDO in inferring potential inputs with limited contextual information, a challenge we plan to address in future work. Another instance of failure is documented in Appendix F.2.

```
┌─────────────────────────────────────────────────────────────────────────────┐
│                              LLM Prediction                                    │
├─────────────────────────────────────────────────────────────────────────────┤
│ I've analyzed the modification patch and the original implementation. Here's   │
│ my assessment:                                                                 │
│                                                                                │
│ <Runtime Errors>                                                               │
│ No potential runtime errors detected.                                          │
│ </Runtime Errors>                                                              │
│                                                                                │
│ <Conclusion> Safe </Conclusion>                                                │
├─────────────────────────────────────────────────────────────────────────────┤
│                              Evaluation log                                     │
├─────────────────────────────────────────────────────────────────────────────┤
│ Traceback (most recent call last):                                             │
│   File "/opt/django__django/django/db/backends/utils.py", line 86, in _execute │
│     return self.cursor.execute(sql, params)                                    │
│   File "/opt/django__django/django/db/backends/sqlite3/base.py", line 396,     │
│      in execute                                                                 │
│     return Database.Cursor.execute(self, query, params)                        │
│ sqlite3.OperationalError: no such column: U0.name                              │
└─────────────────────────────────────────────────────────────────────────────┘
```

Figure 5: Failed example

**How could the detected errors help?**    This section presents a qualitative example illustrating how detected errors can facilitate the correction of flawed patches. We introduce a preliminary patch-fixing algorithm that leverages error messages generated by REDO. Specifically, when presented with a modified patch (referred to as the *original generated patch*) and its associated detected errors, our approach first applies the patch and extracts the modified code chunks, with ten lines of code added before and after each chunk. We then prompt the LLM to refine the code chunks according to the identified errors. Finally, a fixed patch is generated based on the modifications to the code chunks. The detailed LLM prompt is provided in Appendix E.6. For instance, in the case of *Django-12308* utilizing *ACR*, REDO initially identifies a risky attribute call that could potentially lead to an AttributeError. Based on this detection, the patch-fixing algorithm generates an fixed patch that avoids invoking the risky attribute, thereby preventing the AttributeError present in the original patch. A comprehensive description of the algorithm and a detailed analysis of this example are provided in Section G.

## 6    CONCLUSION

In this study, we first present REDO, an innovative error detection framework that operates through a two-step process: differential analysis followed by LLM-based detection. This approach achieves a balanced trade-off between reliability and the scope of detectable errors. We also propose SWE-Bench-Error-Detection (SWEDE), a novel and challenging runtime error detection task that aligns with the increasing deployment of autonomous coding agents responsible for repository-level modifications. Furthermore, we conduct a comprehensive set of quantitative experiments to empirically demonstrate the efficacy of REDO across various tasks. In addition, our qualitative analysis offers insights into the conditions under which LLM integration proves beneficial or falls short, and how detected runtime errors using REDO could help fix the previous flawed patches.

## 7    LIMITATION AND FUTURE WORK

The current implementation of the LLM-based detection step constrains the number of LLM API call on each data point to just one. Expanding the number of API calls, potentially by leveraging agentic AI techniques, could significantly improve error detection capabilities. Additionally, the SWEDE task is presently confined to a binary classification of 'Safe' and 'Unsafe.' As error detection algorithms become more sophisticated, it is crucial to consider expanding the classification schema to encompass a wider spectrum of error types, similar to those addressed in STA. Finally, the advancement of more refined ensemble and patch-fixing algorithms holds the potential to enhance the efficacy of REDO in supporting coding agents.

ETHICS STATEMENT.

Our work leverages results from previous methods, including publicly available sources and the SWE-Bench dataset and leaderboard, as cited in the Experiment section. To the best of our knowledge, this study does not pose any risks related to harmful insights, discrimination, bias, fairness, privacy, or security concerns.

REPRODUCIBILITY STATEMENT.

We provide details of our experiments and implementation in Sections 3 and 5, including the models used and a description of the data processing steps. Given the limited time, we are unable to wrap up all the code files before the submission deadline. However, we will gladly provide them during the rebuttal stage if required. Additionally, for each experiment, we used three random seeds and report both the means and standard deviations.

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

## A   REDO PSEUDOCODE

---

**Algorithm 1** Execution-Free Runtime Error Detection for Coding Agents

---

1: **Input:** original github repo, problem statement $i$, generated patch $p$, function searching method $f$, git apply function $g$, git revert function $r$, static analysis tool $t$, LLM $l$
2: Search for original python script $c$ containing the modified functions
3: Detect runtime errors $S_{\text{Orig}}$ in the original implementation using $t$
4: Apply the patch
5: Detect runtime errors $S_{\text{Mod}}$ in the modified implementation using $t$
6: Revert applied patch
7: **if** $S_{\text{Orig}} \neq S_{\text{Mod}}$ **then**
8:     **return** 'Unsafe'
9: **else**
10:     Prompt l with $i$, original script $c$, and modification patch $p$
11:     **if** Do not detect runtime errors **then**
12:         **return** 'Safe'
13:     **else**
14:         **return** 'Unsafe'
15:     **end if**
16: **end if**

---

## B   CONFUSION MATRICES

## C   ADDITIONAL QUANTITATIVE RESULTS

Table 4: Performance metrics by method and benchmark using Claude-3 OPUS

| Coding agent | Metric | Method | | | | |
|---|---|---|---|---|---|---|
| | | Pyflakes | PyRight | LLM | REDO-Pyflakes | REDO |
| CodeStory | Precision | 32.1 | 43.7 | 43.1 | 35.6 | 39.2 |
| | Recall | 22.8 | 48.1 | 39.2 | 46.8 | 62.0 |
| | F1 | 26.7 | 45.8 | 41.1 | 40.4 | 48.0 |
| Demo | Precision | 34.1 | 33.9 | 34.0 | 33.9 | 33.6 |
| | Recall | 34.9 | 45.3 | 20.9 | 47.7 | 55.8 |
| | F1 | 34.5 | 38.8 | 25.9 | 39.6 | 41.9 |
| Marscode | Precision | 22.9 | 33.8 | 30.8 | 28.7 | 32.8 |
| | Recall | 16.4 | 37.3 | 35.8 | 46.3 | 59.7 |
| | F1 | 19.1 | 35.5 | 33.1 | 35.4 | 42.3 |
| Lingma | Precision | 45.8 | 48.3 | 48.6 | 45.9 | 49.1 |
| | Recall | 27.8 | 29.9 | 35.1 | 51.5 | 54.6 |
| | F1 | 34.6 | 36.9 | 40.7 | 48.5 | 51.7 |
| Droid | Precision | 53.3 | 42.1 | 56.0 | 50.6 | 48.2 |
| | Recall | 22.0 | 14.7 | 25.7 | 40.4 | 36.7 |
| | F1 | 31.2 | 21.8 | 35.2 | 44.9 | 41.7 |
| ACR | Precision | 45.3 | 54.7 | 43.1 | 41.2 | 45.9 |
| | Recall | 23.5 | 34.3 | 27.5 | 41.2 | 49.0 |
| | F1 | 31.0 | 42.2 | 33.5 | 41.2 | 47.4 |

## D   ENSEMBLE PATCHES

For a base agent, we denote the number of instance on SWEDE switching from *failed* to *passed* as $M$; and from *passed* to *failed* as $N$. The performance enhancement $E$ of the base agent is therefore measured by the difference in number between newly passed instances to newly failed instances, or

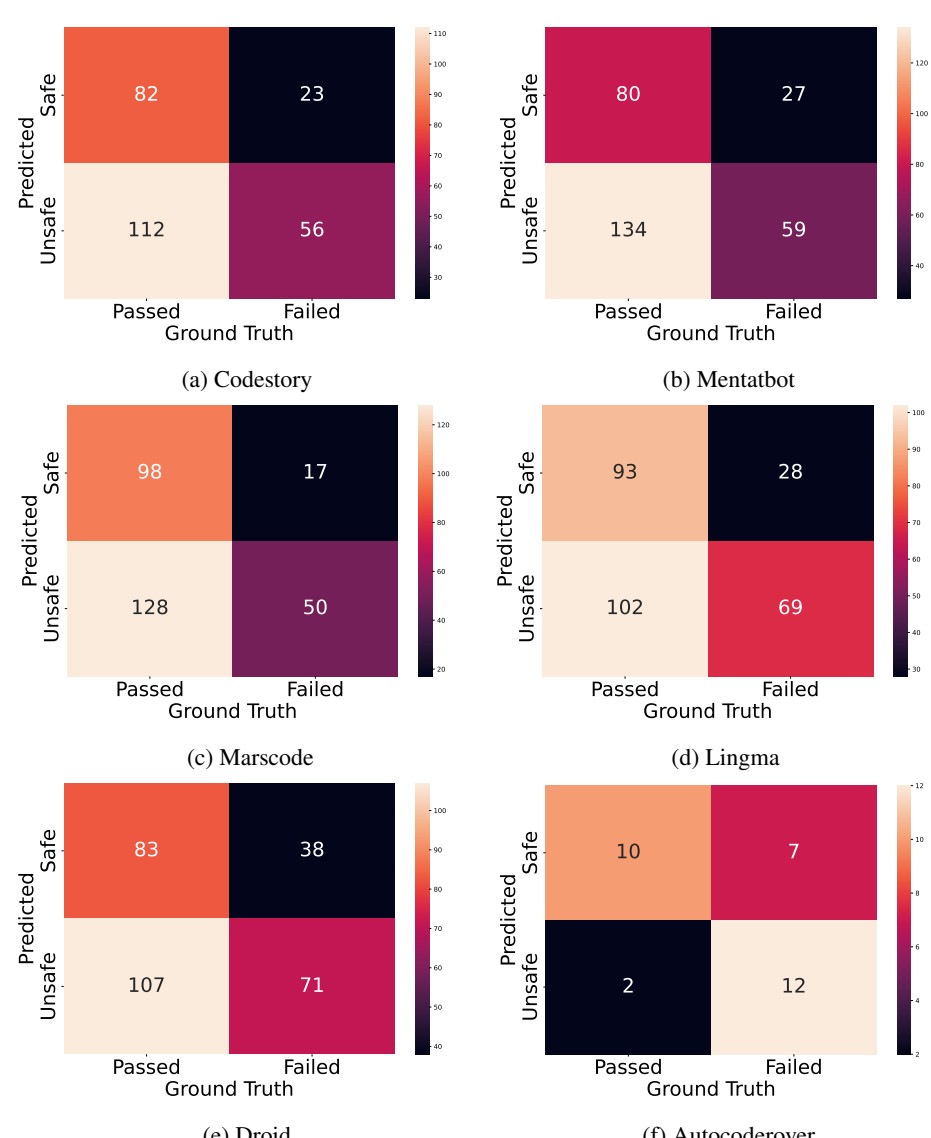

Figure 6: Confusion matrices

$E = M - N$. We further consider two scenarios. First, only passed instances or instances with runtime errors using the base agent are considered; second, all instances are considered. These two scenarios evaluates how well REDO detects specifically runtime errors and any errors, respectively. We report the results in Figure 7a and 7b, where the rows represent different base agents and columns represent auxiliary agents.

First, the enhancements $E$'s are positive for the majority of the entries, and are especially significant when ensemble a base agent with a more powerful auxiliary agent (as shown by numbers in lower triangular entries). The average enhancement is **3.1** in Figure 7a and **4.6** in Figure 7b. **These enhancements are meaningful as more powerful agents usually involve more API calls. Switching to more powerful agents only when necessary can reduce over cost and time.** Second, when the base agent is CodeStory, the enhancements are negative. This could results from the fact that the ensemble algorithm currently does not consider the difference in the coding editing performance between base and auxiliary agents, therefore being over-confident with less powerful auxiliary agents. Since the ensemble algorithm is orthogonal to our contribution, we regard improving the algorithm as an important future work.

Table 5: Performance metrics by method and benchmark using Claude-3 SONNET and temperature=0.5

| Coding agent | Metric | Method | | | | |
|---|---|---|---|---|---|---|
| | | Pyflakes | PyRight | LLM | REDO-Pyflakes | REDO |
| CodeStory | Precision | 32.1 | 43.7 | 33.3 | 32.9 | 34.1 |
| | Recall | 22.8 | 48.1 | 67.1 | 69.6 | 75.9 |
| | F1 | 26.7 | 45.8 | 44.5 | 44.7 | 47.1 |
| Demo | Precision | 34.1 | 33.9 | 31.8 | 31.4 | 32.1 |
| | Recall | 34.9 | 45.3 | 65.1 | 70.9 | 77.9 |
| | F1 | 34.5 | 38.8 | 42.7 | 43.6 | 45.4 |
| Marscode | Precision | 22.9 | 33.8 | 29.5 | 28.2 | 29.3 |
| | Recall | 16.4 | 37.3 | 68.7 | 74.6 | 76.1 |
| | F1 | 19.1 | 35.5 | 41.3 | 41.0 | 42.3 |
| Lingma | Precision | 45.8 | 48.3 | 40.9 | 40.7 | 41.4 |
| | Recall | 27.8 | 29.9 | 69.1 | 74.2 | 74.2 |
| | F1 | 34.6 | 36.9 | 51.3 | 52.6 | 53.1 |
| Droid | Precision | 53.3 | 42.1 | 43.7 | 42.3 | 43.0 |
| | Recall | 22.0 | 14.7 | 67.0 | 70.6 | 70.6 |
| | F1 | 31.2 | 21.8 | 52.9 | 52.9 | 53.5 |
| ACR | Precision | 45.3 | 54.7 | 39.6 | 39.1 | 40.9 |
| | Recall | 23.5 | 34.3 | 63.7 | 68.6 | 72.5 |
| | F1 | 31.0 | 42.2 | 48.9 | 49.8 | 52.3 |

Table 6: Means and standard deviations using PyRight, LLM, and REDO on STA.

| Metric | Method | | |
|---|---|---|---|
| | PyRight | LLM | REDO-PyRight |
| Without context | | | |
| Accuracy | $74.5_{0.1}$ | $61.8_{0.0}$ | $75.1_{0.3}$ |
| Running Accuracy [†] | $76.4_{0.0}$ | $64.1_{0.1}$ | $76.8_{0.3}$ |
| W.F1 | $67.2_{0.1}$ | $54.0_{0.0}$ | $69.9_{0.2}$ |
| Running W.F1 | $70.1_{0.0}$ | $57.2_{0.0}$ | $72.6_{0.2}$ |
| With context | | | |
| Accuracy | $74.5_{0.1}$ | $64.3_{0.1}$ | $77.4_{0.4}$ |
| Running Accuracy | $76.4_{0.0}$ | $66.8_{0.2}$ | $79.3_{0.3}$ |
| W.F1 | $67.2_{0.1}$ | $58.6_{0.1}$ | $74.5_{0.3}$ |
| Running W.F1 | $70.1_{0.0}$ | $62.1_{0.1}$ | $77.5_{0.2}$ |

[†] Metrics with "Running" prefixes are those evaluated on our running results.

# E PROMPT TEMPLATES

## E.1 PROMPT TEMPLATE: SYSTEM PROMPT

```
You are an experienced program analyzer who can identify
    ↪ potential runtime errors without running the programs.
```

## E.2 PROMPT TEMPLATE: ERROR DETECTION ON CODE EDITING

```
A modification patch is proposed to resolve an issue with the
    ↪ current github repo. This modification might introduce
    ↪ runtime errors that cannot be captured by static analysis
```

---

**Algorithm 2** Patch ensemble

---

1: **Input:** base agent $A_{\text{base}}$, auxiliary agent $A_{\text{aux}}$, problem instance $I$, error detection algorithm $D$
2: Generate patch $p$ using base model $A_{\text{base}}$
3: Detect runtime error using REDO
4: **if** No error is detected **then**
5:     **return** Accept $p$
6: **else**
7:     Generate patch $p'$ using auxiliary model $A_{\text{aux}}$
8:     Detect runtime error on $p'$
9:     **if** No error is detected **then**
10:         **return** Accept $p'$
11:     **else**
12:         **return** Accept $p$
13:     **end if**
14: **end if**

---

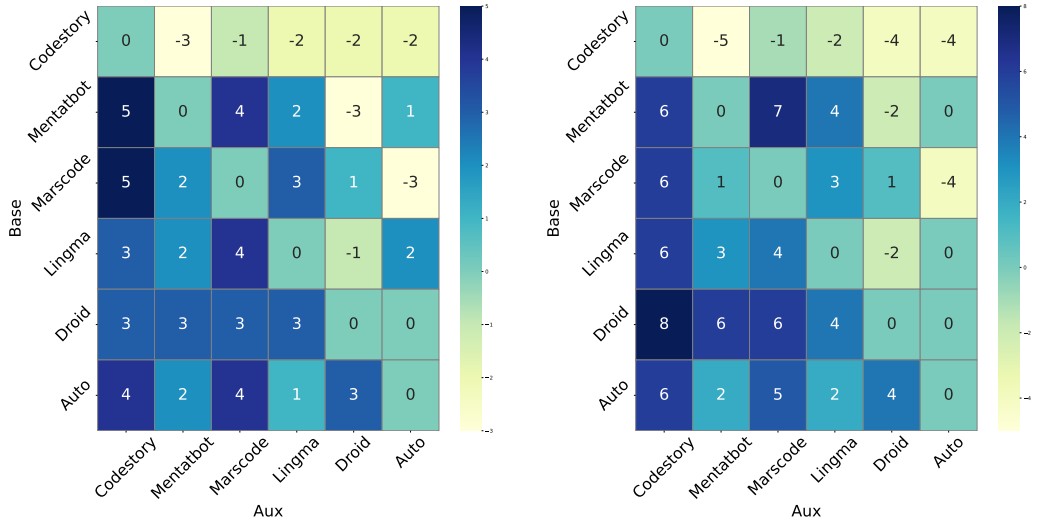

(a) Scenario 1: passed and runtime error instances        (b) Scenario 2: all instances

Figure 7: Patch ensembling using detection results

```
      ↪ tools. Your task is to check whether such runtime errors
      ↪ exist. Typical runtime errors include TypeError,
      ↪ ValueError, AttributeError, and IndexError.

First, you are provided with the problem statement, which
      ↪ describes the issue and hints on how the modification patch
      ↪ will be tested. The problem statement is as follows:
<Problem Statement>
{problem_statement}
</Problem Statement>

Then, you will be provided with the original implementation of
      ↪ python scripts containing modified functions:
<Original Implementation>
{original_implementation}
</Original Implementation>

Finaly, the modification patch is given below:
<Modificatoin Patch>
```

```
{modification_patch}
</Modification Patch>

First, please check if there are potential runtime errors, please
    ↪ list their error type and reasoning in the <Runtime
    ↪ Errors></Runtime Errors> section, with the format
    ↪ [ErrorType]:[Reasoning]. If there are no potential runtime
    ↪ errors, please return 'Safe'; otherwise, please return
    ↪ 'Unsafe'. The conclusion should be wrapped by
    ↪ <Conclusion></Conclusion>.
```

### E.3 PROMPT TEMPLATE: ERROR DETECTION ON CODENET WITH INPUT CONTEXT

```
Given the description of input and the implemented script, please
    ↪ check if the implementation contains runtime errors. You
    ↪ can assume that the inputs are always valid, and relect the
    ↪ common case.

Here is the implementation:
<Implementation>
{implementation}
</Implementation>

Here is the description of the input:
<Input description>
{input}
</Input description>

Potential runtime errors are:
<Error list>
1: 'No Error',
2: 'Other',
3: 'Timeout',
4: 'AssertionError',
5: 'AttributeError',
6: 'decimal',
7: 'EOFError',
8: 'FileNotFoundError',
9: 'ImportError',
10: 'IndentationError',
11: 'IndexError',
12: 'KeyError',
13: 'MathDomainError',
14: 'MemoryError',
15: 'ModuleNotFoundError',
16: 'NameError',
17: 'OSError',
18: 'OverflowError',
19: 're.error',
20: 'RecursionError',
21: 'RuntimeError',
22: 'StopIteration',
23: 'SyntaxError',
24: 'TabError',
25: 'TypeError',
26: 'UnboundLocalError',
27: 'ValueError',
28: 'ZeroDivisionError',
```

```
29: 'numpy.AxisError'
</Error list>

Please explain the logic of the implementation in the
    ↪ "Implementation" section, especially how empty strings or
    ↪ lists are handled. If the implementation is mostly correct
    ↪ and should run without errors in most cases, please claim
    ↪ "No Error"; finally, the index of the identified runtime
    ↪ error that crashes the program in the "Conclusion" section,
    ↪ being wrapped by <Conclusion></Conclusion>.
```

### E.4 PROMPT TEMPLATE: ERROR DETECTION ON CODENET WITHOUT INPUT CONTEXT

```
Given the implemented script, please check if the implementation
    ↪ contains runtime errors. Please assume that the inputs are
    ↪ always valid; and only reflect the most common case.

Here is the implementation:
<Implementation>
{implementation}
</Implementation>

Potential runtime errors are:
<Error list>
1: 'No Error',
2: 'Other',
3: 'Timeout',
4: 'AssertionError',
5: 'AttributeError',
6: 'decimal',
7: 'EOFError',
8: 'FileNotFoundError',
9: 'ImportError',
10: 'IndentationError',
11: 'IndexError',
12: 'KeyError',
13: 'MathDomainError',
14: 'MemoryError',
15: 'ModuleNotFoundError',
16: 'NameError',
17: 'OSError',
18: 'OverflowError',
19: 're.error',
20: 'RecursionError',
21: 'RuntimeError',
22: 'StopIteration',
23: 'SyntaxError',
24: 'TabError',
25: 'TypeError',
26: 'UnboundLocalError',
27: 'ValueError',
28: 'ZeroDivisionError',
29: 'numpy.AxisError'
</Error list>

Please explain the logic of the implementation in the
    ↪ "Implementation" section, especially how empty strings or
    ↪ lists are handled. If the implementation is mostly correct
```

```
972         ↪ and should run without errors in most cases, please claim
973         ↪ "No Error"; finally, the index of the identified runtime
974         ↪ error that crashes the program in the "Conclusion" section,
975         ↪ being wrapped by <Conclusion></Conclusion>.
976
977
```

### E.5    PROMPT TEMPLATE: ERROR DETECTION ON SWE-BENCH-LITE

```
980    A modification patch is proposed to resolve an issue with the
981         ↪ current github repo. This modification might contain
982         ↪ runtime errors that will crash the unit tests but cannot be
983         ↪ captured by static analysis tools. Your task is to check
984         ↪ whether those errors exist in the current modification.
985
986    First, you are provided with the problem statement, which
987         ↪ describes the issue and hints on how the modification patch
988         ↪ will be tested. The problem statement is as follows:
       <Problem Statement>
989    {problem_statement}
990    </Problem Statement>
991
992    Then, you will be provided with the original implementation of
993         ↪ python scripts containing modified functions:
994    <Original Implementation>
       {original_implementation}
995    </Original Implementation>
996
997    Finaly, the modification patch is given below:
998    <Modificatoin Patch>
999    {modification_patch}
1000   </Modification Patch>
1001
1002   Please identify potential runtime errors that can crash the
1003        ↪ program but cannot be captured by static analysis tools;
1004        ↪ and list them in the <Potential Errors></Potential Errors>
1005        ↪ section. Next, Prune errors that are unlikely to be
1006        ↪ relevant to the problem statement. Please list these errors
1007        ↪ in the "Remaining Errors" section, being wrapped by
1008        ↪ <Remaining Errors></Remaining Errors>.
```

### E.6    PROMPT TEMPLATE: PATCH FIXING

```
1012   Your task is to update the provided code files to prevent the
1013        ↪ previously detected runtime errors. You will be provided
1014        ↪ with relevant code chunks and identified errors.
1015
1016   Begin your response by providing a simple smoke test to test the
1017        ↪ updated code within <test></test> tags. The rest of your
1018        ↪ response should provide the updated code to prevent the
1019        ↪ runtime errors, matching the exact format of the provided
1020        ↪  below, including the  and <file> tags, and the
1021        ↪ name and start_line attributes. If a code chunk does not
1022        ↪ need any modification, it can be omitted from your
1023        ↪ response. Each code chunk you update to solve the problem
1024        ↪ must be rewritten in full, including lines that are
1025        ↪ unchanged. The name and start_line XML attributes in your
           ↪ response should always match those in the code below
           ↪ exactly - do not change them. For example, if 100 lines of
```

```
↪ code are passed for a code chunk, but you only modify 5
↪ lines, you must still include the full code chunk in your
↪ response with the original start_line attribute. If you are
↪ able to solve the problem, provide
↪ <outcome>Complete</outcome> in your response, otherwise
↪ provide
<outcome>Incomplete</outcome>, along with brief feedback and next
↪ steps within
<assessment></assessment> tags.

Below is a simple example of a valid response:
<example>
<smoke_test>
from path.to.file import combine_numbers
combine_numbers(123, 456)
</smoke_test>
As requested, in the updated code below, I've rewritten the full
↪ chunks provided, even those parts that remain unchanged,
↪ such as the load_file function.

<file name="path/to/file1.py" start_line="5">
import numpy as np
</file>
<file name="path/to/file1.py" start_line="23">
def load_file(path):
    with open(path, "r") as f:
        content = f.read()
    return content

def combine_numbers(a, b):
    return {{
        "sum": a + b,
        "difference": a - b,
        "product": a * b,
        "quotient": a / b,
        "geometric_mean": geometric_mean(a, b),
    }}
</file>

<outcome>Incomplete</outcome>
<assessment>
Although this patch adds a geometric mean calculation, it does
↪ not import the required function to the file. The next step
↪ is to import the `geometric_mean` function to
↪ path/to/file1.py
</assessment>
</example>

Here are the detected runtime errors:
<Rumetime errors>
{remaining_issues}
</Runtime errors>

Here are the code chunks:

{code}

```

## F ADDITIONAL QUALITATIVE RESULTS

| LLM prediction |
|---|

```
I'll analyze the modification patch for potential runtime errors:
<Runtime Errors>
[AttributeError]: The modification adds `user.email` to the hash value calculation.
However, not all user models in Django are guaranteed to have an email field. If a custom
user model is used that doesn't include an email field, this could raise an AttributeError.
</Runtime Errors>

<Conclusion>
Unsafe
</Conclusion>

The modification introduces a potential runtime error that static analysis tools might not
catch. The main concern is the addition of `user.email` to the hash value calculation.
While many Django user models include an email field, it's not a requirement of the
AbstractBaseUser class. Custom user models might not have this field, which could lead to
an AttributeError at runtime when trying to access the non-existent email attribute.

This change assumes that all user models will have an email field, which isn't always true
in Django applications. To make this change safer, it would be advisable to check for the
existence of the email attribute before using it, or to provide a fallback value if the
email attribute is not present.
```

| Evaluation log |
|---|

```
Traceback (most recent call last):
  File "/opt/pyenv/versions/3.6.15/lib/python3.6/unittest/case.py", line 59, in
testPartExecutor yield
  File "/opt/pyenv/versions/3.6.15/lib/python3.6/unittest/case.py", line 523, in subTest yield
  File "/opt/django__django/tests/auth_tests/test_tokens.py", line 57, in
test_token_with_different_email tk1 = p0.make_token(user)
  File "/opt/django__django/django/contrib/auth/tokens.py", line 28, in make_token return
self._make_token_with_timestamp(user, self._num_seconds(self._now()))
  File "/opt/django__django/django/contrib/auth/tokens.py", line 70, in
_make_token_with_timestamp self._make_hash_value(user, timestamp))
  File "/opt/django__django/django/contrib/auth/tokens.py", line 98, in _make_hash_value
return str(user.pk) + user.password + str(login_timestamp) + user.email + str(timestamp)

AttributeError: 'CustomEmailField' object has no attribute 'email'
```

| Generated patch |
|---|

```
diff --git a/django/contrib/auth/tokens.py b/django/contrib/auth/tokens.py
index 93e2d71af2..6372dd927c 100644
--- a/django/contrib/auth/tokens.py
+++ b/django/contrib/auth/tokens.py
@@ -83,7 +83,8 @@ class PasswordResetTokenGenerator:
        used:
        1. The password field will change upon a password reset (even if the
           same password is chosen, due to password salting).
-       2. The last_login field will usually be updated very shortly after
+       2. The email field will change and invalidate the token.
+       3. The last_login field will usually be updated very shortly after
           a password reset.
        Failing those things, settings.PASSWORD_RESET_TIMEOUT eventually
        invalidates the token.
@@ -94,7 +95,7 @@ class PasswordResetTokenGenerator:
        # Truncate microseconds so that tokens are consistent even if the
        # database doesn't support microseconds.
        login_timestamp = '' if user.last_login is None else
user.last_login.replace(microsecond=0, tzinfo=None)
-       return str(user.pk) + user.password + str(login_timestamp) + str(timestamp)
+       return str(user.pk) + user.password + str(login_timestamp) + user.email +
str(timestamp)

    def _num_seconds(self, dt):
        return int((dt - datetime(2001, 1, 1)).total_seconds())
```

Figure 8: Generated patch and LLM response on Django-13551 using Lingma.

### F.1 ANOTHER SUCCESSFUL EXAMPLE

This example is on instance *Django-10924* using *Lingma*. As shown in Figure 10, the modification
fails due to calling a non-callable object *field.path*. According to the modification patch in Figure 10,

this happens if a non-callable *path* variable is input into the init function. As this error is only triggered during runtime, static analysis tools fail to capture it. On the other hand, as shown in Figure 10, the LLM succeeds to identify this situation and foresee the runtime error.

### F.2 ANOTHER FAILED EXAMPLE

This example is on instance *Sympy-13471* using *Mentabot*, which actually passed the unit test. As shown in Figure 11, the LLM identifies a corner case where the variable *num* is not a string, which will fail the program. However, since this corner case is not triggered, the instance passed the unit test, making the LLM prediction a false positive alarm.

## G  PATCH FIXING

As depicted in Figure 12, REDO identifies two potential runtime errors, including an AttributeError triggered by invoking the prepare_value attribute from JSONField. The evaluation log after executing the modified implementation confirms that this AttributeError indeed caused a crash. Subsequently, only basing on the results from REDO and without knowing the true error, Figure 13 shows that the corrected patch avoids invoking JSONField, which successfully mitigates the AttributeError.

We remark that although the AttributeError is resolved, the fixed patch may introduce new AssertionError instances or other runtime errors. The AssertionError lies beyond the detection capabilities of REDO, and the emergence of additional runtime errors suggests that an iterative process may be required to fully eliminate all such errors.

| LLM prediction |
|---|

I've analyzed the modification patch and the original implementation. Here's my assessment:

<Runtime Errors>
No potential runtime errors detected.
</Runtime Errors>

<Conclusion>
Safe
</Conclusion>

The modification patch appears to be safe and doesn't introduce any obvious runtime errors. The changes are focused on preserving the GROUP BY clause for subqueries, which is a logical extension of the existing functionality. The modifications are made in a way that's consistent with the existing code structure and don't introduce any new variables or complex logic that could lead to runtime errors.

The main changes involve:

1. Adding a check to preserve the GROUP BY clause when building filters with subqueries.
2. Ensuring the GROUP BY clause is preserved when splitting excludes.

These modifications are straightforward and don't alter the fundamental behavior of the query building process. They simply ensure that GROUP BY information is maintained when working with subqueries, which should improve the correctness of the generated SQL without introducing new risks.

| Evaluation log |
|---|

```
Traceback (most recent call last):
  File "/opt/django__django/django/db/backends/utils.py", line 86, in _execute
    return self.cursor.execute(sql, params)
  File "/opt/django__django/django/db/backends/sqlite3/base.py", line 396, in execute
    return Database.Cursor.execute(self, query, params)
sqlite3.OperationalError: no such column: U0.name
```

| Generated patch |
|---|

```diff
diff --git a/django/db/models/fields/__init__.py b/django/db/models/fields/__init__.py
index 0c64748ccd..7efc0967c8 100644
--- a/django/db/models/fields/__init__.py
+++ b/django/db/models/fields/__init__.py
@@ -1663,7 +1663,9 @@ class FilePathField(Field):

     def __init__(self, verbose_name=None, name=None, path='', match=None,
                  recursive=False, allow_files=True, allow_folders=False, **kwargs):
-        self.path, self.match, self.recursive = path, match, recursive
+        self.path_callable = path if callable(path) else None
+        self.path = path() if callable(path) else path
+        self.match, self.recursive = match, recursive
         self.allow_files, self.allow_folders = allow_files, allow_folders
         kwargs.setdefault('max_length', 100)
         super().__init__(verbose_name, name, **kwargs)
@@ -1687,7 +1689,9 @@ class FilePathField(Field):

     def deconstruct(self):
         name, path, args, kwargs = super().deconstruct()
-        if self.path != '':
+        if self.path_callable:
+            kwargs['path'] = self.path_callable
+        elif self.path != '':
             kwargs['path'] = self.path
         if self.match is not None:
             kwargs['match'] = self.match
@@ -1708,8 +1712,9 @@ class FilePathField(Field):
         return str(value)

     def formfield(self, **kwargs):
+        path = self.path_callable() if self.path_callable else self.path
         return super().formfield(**{
-            'path': self.path,
+            'path': path,
            'match': self.match,
            'recursive': self.recursive,
            'form_class': forms.FilePathField,
```

Figure 9: Generated patch and LLM response on Django-11797 using Codestory.

| LLM prediction |
|---|

I've analyzed the modification patch and the original implementation. Here's my assessment:

<Runtime Errors>
[TypeError]: The modification assumes that num[1] is always a string, but it might not be. If num[1] is already an integer (long), calling .rstrip('L') on it will raise a TypeError.
</Runtime Errors>

<Conclusion>Unsafe</Conclusion>

The modification patch introduces a potential runtime error. It assumes that num[1] is always a string and tries to call the .rstrip('L') method on it. However, in the original implementation, num[1] could already be an integer (long) if the input comes from a non-pickled source. In such cases, calling .rstrip('L') on an integer would raise a TypeError.

To make this modification safe, it should first check if num[1] is a string before attempting to strip the 'L' suffix. For example:

```python
if isinstance(num[1], string_types):
    num[1] = num[1].rstrip('L')  # Strip 'L' suffix if present
num[1] = long(num[1], 16)
```

This change would make the modification safe for all input types.

| Evaluation log |
|---|

```
================================================================== ERROR:
test_callable_path (model_fields.test_filepathfield.FilePathFieldTests) ------------------------
--------------------------------------------- Traceback (most recent call last): File
"/opt/django__django/tests/model_fields/test_filepathfield.py", line 21, in test_callable_path
self.assertEqual(field.path(), path) TypeError: 'str' object is not callable
```

| Generated patch |
|---|

```diff
diff --git a/django/db/models/fields/__init__.py b/django/db/models/fields/__init__.py
index 0c64748ccd..7efc0967c8 100644
--- a/django/db/models/fields/__init__.py
+++ b/django/db/models/fields/__init__.py
@@ -1663,7 +1663,9 @@ class FilePathField(Field):

     def __init__(self, verbose_name=None, name=None, path='', match=None,
                  recursive=False, allow_files=True, allow_folders=False, **kwargs):
-        self.path, self.match, self.recursive = path, match, recursive
+        self.path_callable = path if callable(path) else None
+        self.path = path() if callable(path) else path
+        self.match, self.recursive = match, recursive
         self.allow_files, self.allow_folders = allow_files, allow_folders
         kwargs.setdefault('max_length', 100)
         super().__init__(verbose_name, name, **kwargs)
@@ -1687,7 +1689,9 @@ class FilePathField(Field):

     def deconstruct(self):
         name, path, args, kwargs = super().deconstruct()
-        if self.path != '':
+        if self.path_callable:
+            kwargs['path'] = self.path_callable
+        elif self.path != '':
             kwargs['path'] = self.path
         if self.match is not None:
             kwargs['match'] = self.match
@@ -1708,8 +1712,9 @@ class FilePathField(Field):
         return str(value)

     def formfield(self, **kwargs):
+        path = self.path_callable() if self.path_callable else self.path
         return super().formfield(**{
-            'path': self.path,
+            'path': path,
             'match': self.match,
             'recursive': self.recursive,
             'form_class': forms.FilePathField,
```

Figure 10: Generated patch and LLM response on Django-10924 using Mentatbot.

| LLM prediction |
|---|

I've analyzed the modification patch and the original implementation. Here's my assessment:

```
<Runtime Errors>
[TypeError]: The modification assumes that num[1] is always a string, but it might not be.
If num[1] is already an integer (long), calling .rstrip('L') on it will raise a TypeError.
</Runtime Errors>

<Conclusion>Unsafe</Conclusion>
```

The modification patch introduces a potential runtime error. It assumes that num[1] is always a string and tries to call the .rstrip('L') method on it. However, in the original implementation, num[1] could already be an integer (long) if the input comes from a non-pickled source. In such cases, calling .rstrip('L') on an integer would raise a TypeError.

To make this modification safe, it should first check if num[1] is a string before attempting to strip the 'L' suffix. For example:

```python
if isinstance(num[1], string_types):
    num[1] = num[1].rstrip('L')  # Strip 'L' suffix if present
num[1] = long(num[1], 16)
```

This change would make the modification safe for all input types.

| Generated patch |
|---|

```diff
diff --git a/sympy/core/numbers.py b/sympy/core/numbers.py
index 7bc96e68eb..4c8301c6b2 100644
--- a/sympy/core/numbers.py
+++ b/sympy/core/numbers.py
@@ -1042,6 +1042,7 @@ def __new__(cls, num, dps=None, prec=None, precision=None):
                 # it's a hexadecimal (coming from a pickled object)
                 # assume that it is in standard form
                 num = list(num)
+                num[1] = num[1].rstrip('L')  # Strip 'L' suffix if present
                 num[1] = long(num[1], 16)
                 _mpf_ = tuple(num)
             else:
```

Figure 11: Generated patch and LLM response on Sympy-13471 using Mentatbot.

```
┌─────────────────────────────────────────────────────────────────────────┐
│                              LLM prediction                               │
├─────────────────────────────────────────────────────────────────────────┤
│ I've analyzed the modification patch and the original implementation.     │
│ Here's my assessment:                                                     │
│                                                                           │
│ <Runtime Errors>                                                          │
│ [AttributeError]: The patch assumes that JSONField has a 'prepare_value'  │
│ method. If this method doesn't exist on the JSONField class, it could     │
│ raise an AttributeError.                                                  │
│                                                                           │
│ [ImportError]: The patch adds an import statement for JSONField from      │
│ django.db.models. If this import is not available in the version of       │
│ Django being used, it could raise an ImportError.                         │
│ </Runtime Errors>                                                         │
│                                                                           │
│ <Conclusion>Unsafe</Conclusion>                                           │
│                                                                           │
│ The modification patch introduces potential runtime errors that cannot be │
│ captured by static analysis tools. The main concerns are:                 │
│                                                                           │
│ 1. The patch assumes the existence of a 'prepare_value' method on         │
│ JSONField. If this method is not part of the JSONField API, it could lead │
│ to an AttributeError at runtime.                                          │
│                                                                           │
│ 2. The added import statement for JSONField might not be compatible with  │
│ all versions of Django, potentially causing an ImportError.               │
│                                                                           │
│ These issues wouldn't be caught by static analysis tools because they     │
│ depend on the specific implementation and version of Django being used.   │
│ Therefore, it's important to thoroughly test this change across different  │
│ Django versions and ensure that the JSONField API includes the            │
│ 'prepare_value' method before implementing this patch.                    │
├─────────────────────────────────────────────────────────────────────────┤
│                              Evaluation log                               │
├─────────────────────────────────────────────────────────────────────────┤
│ ======================================================================    │
│ ERROR: test_json_display_for_field (admin_utils.tests.UtilsTests)         │
│ [<object object at 0x7f9860dbbe90>] (value={('a', 'b'): 'c'})             │
│ ----------------------------------------------------------------------    │
│ Traceback (most recent call last):                                        │
│   File "/opt/pyenv/versions/3.6.15/lib/python3.6/unittest/case.py",       │
│ line 59, in testPartExecutor                                              │
│     yield                                                                 │
│   File "/opt/pyenv/versions/3.6.15/lib/python3.6/unittest/case.py",       │
│ line 523, in subTest                                                      │
│     yield                                                                 │
│   File "/opt/django__django/tests/admin_utils/tests.py", line 192, in     │
│ test_json_display_for_field                                               │
│     display_for_field(value, models.JSONField(), self.empty_value),       │
│   File "/opt/django__django/django/contrib/admin/utils.py", line 404, in  │
│ display_for_field                                                         │
│     return field.prepare_value(value)                                     │
│ AttributeError: 'JSONField' object has no attribute 'prepare_value'       │
│                                                                           │
│                                                                           │
│ ----------------------------------------------------------------------    │
│ Ran 22 tests in 0.779s                                                    │
│                                                                           │
│ FAILED (errors=4)                                                         │
│ Destroying test database for alias 'default'                              │
│ ('file:memorydb_default?mode=memory&cache=shared')...                     │
│ Destroying test database for alias 'default'                              │
│ ('file:memorydb_default?mode=memory&cache=shared')...                     │
│ Destroying test database for alias 'default'                              │
│ ('file:memorydb_default?mode=memory&cache=shared')...                     │
│                                                                           │
│ [django__django__3.1] [django__django-12308] Return Code: 1              │
│                                                                           │
│ >>>>> Some Tests Failed                                                   │
│ [django__django__3.1] [django__django-12308] Test script run successful  │
└─────────────────────────────────────────────────────────────────────────┘
```

Figure 12: REDO analysis and evaluation log on the fixed patch.

Figure 13: Original generated patch and fixed patch