# OpenReview forum: "REDO: Execution-Free Runtime Error Detection for Coding Agents"
_ICLR.cc/2025/Conference — Submitted to ICLR 2025_

### Official Review · Reviewer_W6to · 2024-10-28

**Soundness:** 3
**Presentation:** 3
**Contribution:** 2
**Rating:** 5
**Confidence:** 4

**Summary:**

The paper presents REDO, a novel approach at predicting runtime errors in Python code patches by first querying traditional static analysis (SA) tools and extending them with LLM predictions if no error is detected. They compare the method with pure SA and pure LLM methods and one recent work on two datasets, one based on real world GitHub projects and one based on standalone scripts, and conclude that such methods are superior based on obtained F1 scores.

**Strengths:**

- The story is coherent and reads in an engaging way and mostly well-written and properly formatted.
- The figures and tables are well sized and easy to read and interpret.
- The proposed method is interesting in that it combines traditional conservative and novel more sensitive tooling, to obtain the best of both worlds.

**Weaknesses:**

I think the paper presents an interesting idea, however I am not convinced that the claimed SOTA results have merit in the sense that it is claimed in the paper, therefore, without further changes I lean towards reject. Based on my current understanding addressing the following points would be sufficient to change my recommendation towards accept.

1) Lack of precision in assessing method prediction in SWEDE

The benchmark SWEDE considers all error types as one for its evaluation, considering coinciding "method predicts some error" and "some error occurred" as a successful prediction (Sec. 4). I think this score is significantly less interesting than the score used by Bieber et. al (2022) in the previous work, i.e. to consider each error _type_ as a separate class, for several reasons.

- The purpose of error prediction is the potential use of the predicted information for subsequent repair of suggested fixes. The more precise such a prediction the more useful it is to whichever tool (i.e. LLM) is to implement such a fix. Predicting an AttributeError for a non-existing method when the actual error is completely unrelated may be unhelpful at best or confusing at worst for subsequent processing. Ideally one would even want to correctly predict the cause of the error, though this is admittedly difficult to assess in an automatic fashion.
- The previous work, which this work compares to, uses exactly such a metric and a comparison might be unfair or misleading.
- Traditional, rule-based static analysis tools usually predict errors (and their sources) precisely. Related guarantees are often important for applications (i.e. see above) and are already violated by extending results with LLM predictions. It would be interesting to see how precise LLMs are, in order to obtain insight into whether further research in this direction will be required.


I suggest for these reason that either
- the authors provide such a more detailed score for SWEDE or
- the authors present a compelling experiment to investigate whether the predicted errors are already helpful for repair (i.e. they could also still be helpful because they rarely point to entirely unrelated errors, even when not exactly matching)


2) No attempt at generalization of the proposed method

The authors compare the combined performance of static analysis tools and LLMs with preceding work that trains a tailored model on error prediction. Since the proposed method of combining i) precise but conservative static analysis with  ii) an imprecise but more sensitive learned model, is very general, I am wondering a) if it is not possible to combine the preceding work by Bieber et. al. with static analysis in such a way and b) what the performance of such an approach would be.

Consequently it is also not clear whether the improvement in the results stems from the involvement of LLMs or from the clever combination with static analysis tools. A more rigorous analysis of the involved factors would be interesting.

3) Presentation

Overall, the paper could benefit from a number of improvements in the general presentation. Below I present a list of smaller concerns regarding the form:

- In Figure 1 the chart is zoomed in a lot, to highlight results varying around 12% (min-max plotted value). Since the percentages are rather small I would like to see error bars that help assessing the significance of the results.
- Section 4 analyses external dependencies of python files in SWE-Bench "when only one directory level above the location where the modified files resides" is considered. It is not very clear to me what this means and even less so why the dependencies are not calculated for the entire repository (instead of vaguely mentioning that it "could become even more intimidating").
- Algorithm 1 is referenced in Sec. 3.2. Since it was not mentioned that the algorithm is in the appendix, it was difficult to find. I recommend either including the algorithm in the main body (which might also help general understanding or mentioning this) or explicitly referring to "Algorithm 1 in the Appendix"
- I don’t understand why the bolded text in the "STA dataset" section of Sec. 5.1 is bolded and which "different conclusions" it is referring to.
- Sec 5.1. is generally a large swath of text that might benefit from refactoring (i.e. moving details to the appendix and highlighting the main points more easily digestible)
- Table 3 would better fit on page 7 together with the corresponding dataset paragraph
- (A shortened version of) the analyzed patch and code for figures 4 and 5 would aid its understanding. The respective evaluation log could correspondingly be shortened.
- Appendix Section B is apparently completely empty.
- Row spacing in several tables is quite tight, resulting in intersections between presented numbers, underlines and table lines.
- Somehow no recall is provided for the CodeNet dataset and overall very different types of numbers are presented while the datasets do not fundamentally seem different. Similar metrics would IMO aid understanding here.

**Questions:**

- (Referring to W1) Can you provide a more detailed score on how precise LLM predictions are when differentiating between error types? Or alternatively provide some insight on why such differentiation is not needed?
- (Referring to W2) Could you provide some insight on the possibility and impact of combining static analysis with the method proposed by Bieber et. al.?
- What are the error margins in Figure 1?
- What does the bolded text in "STA dataset" refer to (see Nitpicks)?
- What is the expected impact of applying the proposed method in languages with stronger type safety guarantees? Or put differently, which kind of errors is most reliably predicted by static analysis tools and LLMs respectively and what is the expected impact when errors caught by type systems are covered by the compiler?
- Is there a reason for the different metrics of CodeNet and SWEDE and can the assessments be unified in their metrics?

---

> ### Author Response · Authors · 2024-11-21
>
> ## Lack of precision in assessing method prediction in SWEDE
> We acknowledge that the binary classification-like analysis is relatively limited. The decision to label each instance as 'safe' or 'unsafe,' rather than by specific runtime error types, stems from the fact that we can only observe runtime errors that fail the unit tests. However, other runtime errors induced by the patch might remain undetected if not triggered. Consequently, labeling instances by their error types would result in noisy labels, making it difficult to fairly evaluate the error detection algorithms.
>
> To further explore the utility of the predicted errors, we refer to the patch ensemble results (link: [https://ibb.co/McZh0Kr](https://ibb.co/McZh0Kr) and also reported in Appendix D and discussed in Section 5.1) and the qualitative examples in Appendix G. The patch ensemble results demonstrate that leveraging REDO's detection outcomes allows us to ensemble patches from different coding agents and generally improve overall performance. Additionally, the qualitative examples in Appendix G illustrate how LLM reasoning about detected runtime errors can guide coding agents to refine patches and resolve runtime errors.
>
> Lastly, we note that we predicted execution errors based solely on the outputs of existing coding agents, without actually running the agents, as most state-of-the-art (SOTA) coding agents are proprietary. By directly analyzing their results, we could evaluate REDO's performance across a wide spectrum of methods.
>
> ## No attempt at generalization of the proposed method
> First, we thank the reviewer for acknowledging the generality of our approach, which combines a precise but conservative method with an imprecise yet more sensitive learning-based model. We would like to point out that the method proposed by Bieber et al. (2022) is limited in the context of coding agents, as their model only takes generated code as input and outputs an error type. It lacks the ability to comprehend the runtime context or provide reasoning for its detection.
>
> In contrast, LLMs can incorporate additional information, such as the problem statement and the original implementation, while also generating reasoning for their detection. As demonstrated in Appendix G, this reasoning can assist coding agents in refining their current patches and addressing runtime errors more effectively.
>
> ## Presentation
> We thank the reviewer for providing constructive suggestions on improving the presentation. We will incorporate these recommendations into our revision.
>
> ## Questions
> - Please refer to the response to W1.
> - Please refer to the response to W2.
> - We reported the corresponding error margins in Table 6 and will include them in Figure 1 in our revision.
> - We apologize for the confusion. The STA dataset refers to the dataset used in Bieber et al. (2022), which is built upon CodeNet.
> - In general, we found static analysis tools to be most effective in detecting errors where function or variable types can be inferred. This insight motivated us to use Pyright, which excels at type inference, as our static analysis tool. LLMs, on the other hand, are particularly helpful when the running context is unclear, making the types of related variables or functions uncertain. In such cases, LLMs can reason about potential running contexts and detect possible runtime errors.
> Furthermore, for languages with strong type safety guarantees, we expect static analysis tools to perform significantly better.
> - For SWEDE, we consider the label noise discussed in response to W1. For metrics on CodeNet, we followed the approach in Bieber et al. (2022) to ensure a fair comparison.

---

> ### Comment · Reviewer_W6to · 2024-11-21
>
> I thank the authors for the responses. However, the repsonses are overall unsatisfactory, since they largely fail to address the core concern or even raise further questions.
>
> ### Lack of precision
>
> The authors decide to not provide a score for the in-class precision of their method, and instead argue that no further differentiation is needed as i) precise errors types would be too noisy and ii) the detected errors still help agents at improving the results.
>
> i) sounds like pure speculation to me, to consolidate this claim I would expect some (manual?) analysis on whether reported errors of LLMs are actually present and how often these are not triggered by unit tests. The final metric also only compares to triggered errors by unit tests - if the model reports an error that is not the cause of the failure, it is not at all clear whether the error was correctly predicted.
>
> ii) The experiments in Appendix G do precisely not involve any information flow from the detected error to the agent, beyond the classification as safe/unsafe. The models developed by Bieber et. al. or static-analysis-only error detection provide the same necessary information for the provided experiment. So indeed this part somewhat justifies that "unsafe/safe" is a useful-enough classification, but it is unclear if the proposed method is superior in this setting (i.e. instead of F1 scores, for this setting precision or recall might be more important, additionally derived details could provide crucial insight to aid agents in repairing the previous patch).
>
> ### No generalized / fair comparison
>
> It is not clear at all from the provided experiments that the reasoning provided by the LLM is actually helpful to the agents in any way (refer to ii) of the previous answer). Moreover Table 3 only compares F1 scores - the additional reasoning is not relevant for this metric. That LLMs can make better use of the surrounding context to achiever overall higher F1 score is not based on these results.

---

> > ### Author Response · Authors · 2024-11-21
> >
> > We thank the reviewer for the prompt response. And we would like to clarify some unclear points:
> >
> > ## 1, Lack of precision
> >
> > First, we emphasize that besides our algorithm REDO, the issue we address—evaluating key components in coding agent pipelines—is of significant value to the academic community. Our work could inspire further research in this domain, ultimately contributing to the development of more effective and robust tools for coding agents.
> >
> > Next, we thank the reviewer for pointing out the limitations with the evaluation metrics. We agree that providing stats on the error type precision would be informative for future readers. Although we cannot finish that experiment by the rebuttal deadline, we commit to provide those information in the revised version.
> >
> > Furthermore, regarding the current metrics, we wish to clarify the following points:
> >
> > - For the missing manual analysis, we refer the reviewer to the qualitative analysis in Appendix F. Here, we demonstrate that the predicted runtime errors from the LLM are consistent with the evaluation results post-execution.
> > - We acknowledge that the final metric primarily considers errors triggered by unit tests and that the errors reported by the model may differ from the actual triggered errors. This limitation motivated the patch ensemble experiment, which evaluates how much REDO improves performance regardless of error type. However, we agree that incorporating a metric at the error type level—similar to those used in Bieber et al. (2022)—would be insightful and will include this in the revised version.
> >
> > ii) We are sorry for any confusion with the information fed into patch fixing and direct the reviewer to Appendix E.6, which details the prompt used to fix problematic patches based on previous detection results. In this prompt, we include the *remaining_issues*, which list the error type and reasoning. For example:
> >
> > **[AttributeError]: The modification adds user.email to the hash value calculation. However, not all user models in Django are guaranteed to have an email field. If a custom user model is used that doesn't include an email field, this could raise an AttributeError.**
> >
> > This additional reasoning information helps the LLM understand why a runtime error is detected and provides hints for resolving it.
> >
> > We note that such reasoning information is absent in previous discriminative models, which only output predictions without explanations.
> >
> > ## 2, No generalized/ fair comparison
> > For concerns about reasoning, please refer to the explanation above.
> >
> > Regarding F1 scores, we direct the reviewer to Appendix E.2, where we outline the LLM prompt for SWEDE tasks. The prompt includes the *original_implementation*, providing important context, such as the typical usage of variables and functions. In our experiments, different prompts were tested, and the one including the *original_implementation* yielded the best F1 scores.
> >
> > The ablation study comparing prompts with and without the *original_implementation* helped us better understand how LLMs leverage surrounding context to improve performance. We will include these results in the revised version.
> >
> > Overall, we sincerely thank the reviewer for the constructive suggestions on our submission. We are glad to address any additional questions or concerns you may have.

---

> ### Comment · Reviewer_W6to · 2024-11-22
>
> I acknowledge that the authors want to contribute to a relevant field, concretely to more closely analyze how error detection in LLM Agents can be improved to correct patches swiftly. However, a meaningful contribution would require more thorough analysis than what is presented in the paper.
>
> - The example from Appendix F is a single example that could well be cherry picked. It further does not feature an instance where the LLM correctly predicts an error that is not triggered by the unit tests, which is what I am concerned about. A manual investigation of several (at least 20), randomly drawn samples would be required to alleviate this concern.
> - I am sorry about the confusion regarding App. G. The prompt from App. E.6 appears to be the prompt used for the single example presented in App. G and uses the additional reasoning by the LLM. However, this is again a single example and could be cherry-picked. The quantitative experiments meant to corroborate that REDO is useful for patch fixing (Ensemble Patch, App. D,  Algorithm 2) do not appear to use this prompt or any reasoning generated by REDO.
>
> Further I highly recommend the authors to present results as requested.
>
> 1) What is the performance of Bieber et. al. combined with PyRight on the STA dataset?
> 2) How likely is it that the LLM predicts an error that is present in the code but not triggered by unit tests (manual analysis of 20 samples)? or alternatively: What is the performance of the LLM in predicting precisely one of the error types triggered in the unit tests?
> 3) What is the performance of other approaches (Bieber et al, PyRight, Bieber + PyRight) in the "Ensemble Patching" setting?
>
> The previous discussion did not address any of my raised concerns, either unintentionally or intentionally. I do recommend the authors to not answer unless they can provide concrete numbers, since the content of the paper is not sufficient to alleviate my concerns. These are mainly, that LLM + PyRight is not superior to e.g. Bieber et. al. + PyRight, and that REDO is not quantitatively more helpful for guiding Agents in patch correction than only Static Analysis or Bieber et. al..

---

### Official Review · Reviewer_mtjz · 2024-10-30

**Soundness:** 2
**Presentation:** 2
**Contribution:** 2
**Rating:** 3
**Confidence:** 3

**Summary:**

The paper introduces REDO, a framework for error detection with large language modes (LLMs) without executing code. REDO combines static analysis with LLM-based detection to capture errors.

- The static analysis tools (e.g., PyRight) detects runtime errors by comparing the error profile of the original code with that of the modified code. By focusing on new errors introduced by modifications, this approach filters out false positives related to the original code and flags modifications that introduce potential runtime issues, such as SyntaxErrors or AttributeErrors.

- For instances where no errors are found through static analysis, an LLM (like Claude) is prompted with the problem context, original code, and modified patch to reason about runtime errors that might depend on specific inputs or complex conditions. This step helps detect context-sensitive errors, such as TypeErrors, that static analysis might miss, improving error detection without executing the code.


Additionally, the paper presents the SWEDE benchmark, an extension of SWE-Bench, to evaluate error detection capabilities in repository-level tasks with complex dependencies.

In the experiments, REDO demonstrates improved accuracy and F1 scores compared to baseline methods, but it shows a reduction in precision.

**Strengths:**

- Originality

REDO combines static analysis with LLMs, and although similar approaches are becoming more common, this implementation remains useful.

- Quality

The empirical evaluation of REDO  provides meaningful insights into the effectiveness of the proposed approach.


- Clarity

Overall, the paper is not difficult to follow.

- Significance

REDO addresses an important issue by ensuring program code security with LLMs.

**Weaknesses:**

- Overclaim

This paper claims to focus on error detection for coding agents; however, the REDO approach appears to be a more broadly applicable method for general program code error detection, which creates a misalignment in the work's focus.

- Limited Novelty

REDO combines PyRight with LLMs in a relatively straightforward way, with the LLM engaged only when PyRight deems the code safe. There seems to be no interaction or integration between the static analysis tool and the LLM beyond this sequential setup. In Figure 2, while an arrow suggests information flow from PyRight to the LLM, it is unclear what, if any, information is being transferred. Furthermore, other works also use LLMs alongside static analysis and various other tools, further undermining the novelty of REDO's design.

- Inconclusive Performance Improvement

The LLM in REDO is intended to catch potential errors that PyRight might miss; however, LLMs can sometimes mistakenly flag safe code as erroneous, thereby reducing precision. This limitation is reflected in REDO’s lower precision scores in the experiments, indicating that the addition of LLMs does introduce new issues, instead of consistently improve error detection performance.

**Questions:**

1. Could the authors clarify whether REDO is uniquely designed for coding agents or applicable to general program code error detection?

2. What specific information, if any, is transferred between PyRight and the LLM as indicated by the arrow in Figure 2?

3. Given that LLMs may introduce false positives, what measures are taken to control for this, especially when PyRight has deemed code safe?

---

> ### Author Response · Authors · 2024-11-21
>
> ## 1 Overclaim
> We thank the reviewer for highlighting the broad applicability of our method. While our primary focus is on coding agents, we agree that REDO has potential applications in other scenarios, such as generating static code reviews.
>
> ## 2 Limited Novelty
> The core challenge of REDO lies in balancing the high reliability of static analysis tools with the broader detection scope of LLMs. To address this, we deliberately designed the differential analysis algorithm for static analysis tools to minimize false alarms and crafted prompts for the LLM to detect potential runtime errors more aggressively, thereby maximizing recall. While no explicit information is exchanged between the static analysis tool and the LLM, these design choices collectively optimize the trade-off between reliability and detection scope.
>
> ## 3 Inconclusive Performance Improvement
> We acknowledge that false alarms may still occur with REDO, particularly in cases where the LLM overanalyzes and identifies runtime errors that are unlikely to happen. However, we emphasize that the experimental results in Tables 2 and 3 demonstrate that REDO effectively leverages both static analysis tools and LLMs to achieve improved F1 and weighted F1 scores, indicating enhanced error detection performance. Additionally, the patch ensemble experiments show that by using REDO to ensemble patches from different coding agents, we can generally improve the overall performance of the coding agents.
>
> ## Questions
> - Please refer to the response under "Overclaim."
> - Please refer to the response under "Limited Novelty."
> - We would like to emphasize that when designing the prompt for the LLM, we specifically instruct it to identify all potential runtime errors, thereby maximizing its recall. This approach is intended to address the limitations of static analysis tools, which may fail to detect certain runtime errors.

---

### Official Review · Reviewer_5cnY · 2024-11-02

**Soundness:** 3
**Presentation:** 3
**Contribution:** 2
**Rating:** 3
**Confidence:** 4

**Summary:**

This paper studies how to apply LLM to runtime error detection of source codes that are generated by coding agents. The authors employ a hybrid approach in which a static analysis approach collaboratively works with LLM to detect runtime errors. Specifically, PyRight and Pyflakes are used to detect syntactical errors such as SyntaxError, NameError, while LLMs are used to detect runtime errors in codes. The authors claim that their hybrid approach is execution-free in that no execution is needed to detect such errors. A prototype is implemented and evaluated on a self-designed benchmark SWEDE. The experimental results show the outperformance of the proposed approach REDO over other related techniques.

**Strengths:**

1. important problem. Coding agents face a very big problem of generating buggy and unreadable codes.  Automatically generated codes are difficult to debug manually. It is important to find automatic ways of analyzing LLM-generated codes. The authors provide an alternative solution in this direction. The proposed hybrid approach is reasonable and intuitive. Undoubtedly, LLMs can help to some extent find bugs in programs and could be leveraged with other algorithm-based approaches such as static analysis. This direction deserves further investigation.


2. a clear and simple solution. The proposed approach is clear and easy to understand. The mechanism is simple and easy to implement. That might can be used to solve practical problems, as shown in the experiments. Errors can be found efficiently. As an auxiliary functionality, the approach could be integrated like grammar checkers or syntactic checkers for coding agents, providing useful suggestions and alerts to programmers to be careful of potential bugs that are detected.

**Weaknesses:**

1. The capability of detecting bugs purely relies on backend LLMs and static analysis tools. I didn't see how the two approaches can be deeply integrated and collaborated. Apparently, the simple loose coupling of static analysis tools and LLMs cannot find all bugs in codes. Actually, no method exists to find all bugs. It would be very important to investigate collaborative ways of leveraging the advantages of tools and LLMs to find bugs or errors as many as possible. The current integration approach is very preliminary and shall be further explored.

2. The use of static analysis tools is quite limited. The authors emphasize that their approach is execution-free. I didn't see why we have to consider execution-free. Are there some specific reasons or scenarios we have performance execution-free analysis? In the paper, the authors mentioned that static analysis cannot find input-dependent errors (or struggle to find them in their words). That is not the truth. There are various static analysis approaches to detect input-dependent errors such as symbolic execution, conclic testing. Why not apply these sophisticated approaches to LLM-generated codes?

3. Technical contributions are limited. This point might be related to the above two issues. Throughout the paper, I didn't see any technical contributions, which I think is important to a research paper. To me, the improvement by REDO is not surprising, when comparing it with the purely static analysis tool PyRight. LLMs are extraordinarily powerful.  However, imposing LLMs on PyRight only gives very limited improvement (See Table 3). I believe LLMs can do more for this task if the collaboration between the tool and LLM can be tighter.

**Questions:**

1. What are the benefits of being execution-free? Static analysis is not necessarily execution-free. For instance, symbolic execution is a very powerful static analysis technique. It can be used to detect input-dependent errors.

2. Can LLMs work more tightly with static analysis tools for this task, for instance, LLM-based static analysis or static analysis for LLMs?

3. What are the key challenges that are solved by the proposed approach? Dubbing LLM-generated codes is generally a very challenging task. It could be valuable to identify what these challenges are and how they could be solved separately.

---

> ### Author Response · Authors · 2024-11-21
>
> ## 1 The integration of static analysis tools with LLM
> We note that the core challenge of REDO lies in balancing the high reliability of static analysis tools with the broader detection scope of LLMs. To address this, we deliberately designed the differential analysis algorithm for the static analysis tool to minimize false alarms and crafted prompts for the LLM to aggressively detect potential runtime errors, thereby maximizing recall. While no direct information is passed from the static analysis tool to the LLM, these designs collectively optimize the trade-off between reliability and detection scope.
>
> ## 2 Static analysis tool limitation
> First, we agree that more sophisticated tools could further enhance performance. However, there are two key motivations for using relatively simple static analysis tools like Pyright.
> - Lack of unit tests: Unit tests for the generated code may not always be available. For example, in SWE-Bench, unit tests are inaccessible to coding agents.
> - Efficiency and cost-effectiveness: Since REDO is designed as a plug-and-play tool for coding agents, minimizing its runtime and cost is a priority. To this end, we selected Pyright, which is well-known for its fast runtime, and limited the LLM to a single prompt to reduce API costs.
> ## 3 Technical contributions
> We acknowledge that our algorithm is straightforward. However, we emphasize that augmenting program analysis tools to enhance both their reliability and capabilities is a novel approach that could inspire solutions to other challenges, such as security and privacy.
> Moreover, while we agree that more sophisticated integrations of LLMs and static analysis tools could further improve performance, our current approach already demonstrates significant improvements to static analysis tools, as evidenced in Table 2.
>
> ## Question:
> 1. What are the benefits of being execution-free?
> While we agree that execution feedback is often informative, executing generated code introduces several challenges:
> - Lack of unit tests: Unit tests for the generated code may not always exist. For example, in SWE-Bench, unit tests are inaccessible to coding agents. Even if coding agents generate unit tests, as shown in [1], these tests may not accurately represent the practical runtime context of the generated code.
> - Legal, privacy, and cost concerns: As discussed in [2], executing code can raise legal, privacy, and security concerns, particularly in real-world applications like cloud services. Additionally, execution-based error detection is costly, as only one error can be identified per run; the program halts at the first encountered error. In contrast, REDO can detect multiple potential runtime errors in a single analysis.
> - Complexity of environment setup: Setting up the execution environment can be challenging, often requiring dependency installations and significant computational resources. Moreover, an execution-free algorithm is easier to integrate into state-of-the-art coding assistants, such as Copilot, Amazon Q, or Cursor, which lack the capability to execute the underlying codebase.
> By being execution-free, REDO circumvents these challenges while remaining efficient and scalable.
>
> [1] N Mundler, et al. Code Agents are State of the Art Software Testers, 2024.
> [2] M Lacoste, V Lefebvre. Trusted execution environment for telecoms: Strengths, weakness, opportunities, and threats, IEEE Security & Privacy, 2023.
>
> 2, Can LLMs work more tightly with static analysis tools for this task, for instance, LLM-based static analysis or static analysis for LLMs?
>
> We would like to highlight that in our current algorithm, the differential analysis of static tools and the prompt design for the LLM are carefully crafted to maximize the strengths of both tools and achieve an optimal balance between the high reliability of static analysis tools and the broader detection capabilities of LLMs.
> Since REDO is designed as a tool for coding agents, the algorithm is optimized to detect runtime errors efficiently, avoiding long delays or high costs. However, we acknowledge that if more time and resources were available, a deeper integration of these tools could further enhance performance. For instance, one potential approach could involve using the LLM as an agent capable of multi-turn reasoning to analyze and interpret the outputs from static tools.
>
> 3, What are the key challenges?
> The core challenges are as follows:
> - Integrating static analysis tools and LLMs effectively: Achieving a balance that combines the high reliability of static analysis tools with the broader detection scope of LLMs.
> - Ensuring efficient detection performance: Maintaining strong detection capabilities while minimizing runtime delays and reducing API costs.
> - Developing a fair evaluation framework: Designing a fair and robust method to assess the performance of error detection algorithms on state-of-the-art (SOTA) coding agents.

---

> > ### Comment · Reviewer_5cnY · 2024-11-23
> >
> > I agree that augmenting program analysis tools to enhance both their reliability and capabilities is a practical approach to LLM-generated codes. That is a straightforward solution. It is a methodology rather than an approach. Identifying the challenges and providing solutions are worth investigating. The present work is a little bit shallow. It is widely accepted that LLMs are solution predictors while other algorithmic approaches are solutions validators. How to integrate them more effectively? It is a very important question in almost all the fields that will explore LLMs for problem solving. Code generation is such an instance. I'm expecting more sophisticated solutions that could address essential problems in this task.

---

### Official Review · Reviewer_eYV9 · 2024-11-03

**Soundness:** 3
**Presentation:** 3
**Contribution:** 1
**Rating:** 3
**Confidence:** 4

**Summary:**

This paper introduces REDO which is a system that attempts to detect runtime errors in code modifications without execution by combining static analysis tools with LLMs.

**Strengths:**

I think the paper and the overarching idea of using existing program analysis tools in conjunction with LLMs within agents/systems is quite timely. Several recent breakthroughs in tasks such as SWEBench have been made by designing and integrating several classical tools such as linters, type checkers, repository dependency graphs, etc. I also found the data recipe to create the SWEDE benchmark from existing agent failures to be quite interesting.

**Weaknesses:**

**Motivations unclear**. The paper starts of with SWEBench and software agents as the motivation. However the task suddenly switches to simply predicting execution errors. The paper never circles back on how this prediction could actually help agent/systems that generate code. This makes the overall motivation of the paper quite weak in my opinion. While it is clear that setting up execution-environments for real-world code is challenging and an execution-free analyzer helps, the paper does not show any results that motivate that hypothesis.

**Execution feedback will almost always be better than REDO**. Adding to the point above, assume you have an execution environment for a repository. For most cases, the error messages from execution would contain a lot more information (traceback, type of error, etc.) than the assessment that REDO can provide. Furthermore you can even use an LLM to analyze the failed execution trace and code to produce even better feedback for agents. In that case, I'm wondering if one even needs a system like REDO if the motivation is to integrate with software agents.

**SWEDE an afterthought?**. I am not sure of the contributions of SWEDE as a benchmark as currently presented. While letting models figure out execution-based errors before execution takes place seems quite important and powerful, the current analysis or results do not explore this at all. Further more the benchmark reduces complex error detection and root cause analysis to binary classification. With that it seems like SWEDE is a means to evaluate REDO but it's unclear the value beyond that. There's potential for future versions of SWEDE to be a more comprehensive benchmark around predicting execution results that tap into whether or not LLMs truly understand code.

**Questions:**

Que: Could the authors clarify the motivations of the paper beyond integration with agents, if any? That would maybe help view the current set of results with more context.

---

> ### Author Response · Authors · 2024-11-21
>
> ## 1, How does REDO help coding agents?
> We would like to highlight the patch ensemble results ([https://ibb.co/McZh0Kr](https://ibb.co/McZh0Kr)  and reported in Appendix D and referenced in Section 5.1, as well as summarized below) and the qualitative examples in Appendix G. The patch ensemble results demonstrate that by leveraging the detection outcomes from REDO, we can effectively combine patches generated by different coding agents to enhance overall performance. As shown in the qualitative examples in Appendix G, the reasoning provided by the LLM for detected runtime errors can guide the coding agent in refining patches and resolving runtime errors.
> Additionally, we only predicted execution errors on the outputs of existing coding agents, without directly running these agents, as most state-of-the-art coding agents are proprietary. By directly analyzing their outputs, we were able to evaluate the effectiveness of REDO across a diverse range of methods.
>
> ## 2, Motivation for execution-free runtime error detection
> First, we agree that execution feedback is often informative; however, executing generated code comes with several challenges:
> - Lack of unit tests: Unit tests for the generated code may not always be available. For instance, in SWE-Bench, unit tests are inaccessible to coding agents. Even if coding agents can generate unit tests, as shown in [1], these tests might fail to reflect the practical execution context of the generated code.
> - Legal, privacy, and cost concerns: As discussed in [2], executing code can raise legal, privacy, or security issues in real-world scenarios, such as in cloud services. Additionally, execution-based error detection can be costly since only one error is identified per run, as the program halts upon encountering the first error. In contrast, REDO can identify multiple potential runtime errors in a single pass.
> - Environment setup challenges: Setting up the execution environment can be complex, often requiring dependency installations and consuming significant computational resources. Moreover, an execution-free algorithm like REDO can be more easily integrated into state-of-the-art coding assistants such as Copilot, Amazon Q, or Cursor, which do not execute the underlying codebase."
>
> [1] N Mundler, et al. Code Agents are State of the Art Software Testers, 2024.
> [2] M Lacoste, V Lefebvre. Trusted execution environment for telecoms: Strengths, weakness, opportunities, and threats, IEEE Security & Privacy, 2023.
>
> ## 3 SWEDE an afterthought?
> We note that REDO can be seamlessly used as a plug-and-play tool for coding agents. As mentioned, the primary motivation for proposing SWEDE is that most state-of-the-art (SOTA) methods are proprietary. Consequently, to evaluate error detection algorithms on these SOTA agents, we must rely on the output from these agents rather than directly running them.
> We acknowledge that the binary classification-like analysis is relatively limited. The decision to label each instance as “safe” or “unsafe,” rather than categorizing specific runtime error types, stems from the observation that we can only detect runtime errors that fail the unit tests. However, other runtime errors introduced by the patch might remain undetected because they were not triggered during execution. As a result, labeling instances by error types could produce noisy labels, leading to an unfair evaluation of error detection algorithms.
> Finally, we emphasize that in addition to the binary classification-like analysis, we also propose the patch ensemble metric, which offers a practical means to assess how error detection algorithms enhance coding agent performance. We believe that combining the binary classification-like metrics with the patch ensemble evaluation provides a more comprehensive and fair assessment of error detection algorithms.
>
> ## To Question:
> While we emphasize that REDO is primarily designed for coding agents, as noted by reviewer mtjz, it can also be applied to other tasks, such as code generation. Another potential application is leveraging LLM feedback to automatically generate static code reviews.

---

> > ### Comment · Reviewer_eYV9 · 2024-11-27
> >
> > Thanks for your response.
> >
> > > motivation being "lack of unit tests" and "environment setup"
> >
> > while I agree these are challenges for execution, REDO does not solve these challenges while still providing the benefits of execution feedback. If the sole motivation is integration with agents to provide feedback to improve code, the feedback needs to be of the same quality as that provided by execution (tracebacks, error messages, etc.) -- I just don't see REDO doing any of that. REDO is just an application of a more complex linter (which as I mentioned has already been integrated into several agents). Furthermore even the linter used is off-the-shelf and not a contribution of this paper.
> >
> > Finally, it is still super unclear to me what the paper proposes and how the claim that it helps LLMs for code or even agents is true--which is the critical part for a contribution here. The authors mention `As shown in the qualitative examples ... the reasoning provided by the LLM for detected runtime errors can guide the coding agent in refining patches and resolving runtime errors` in the response. I don't think 1/2 qualitative examples are enough to make the claim that REDO helps coding agents. As a result, I would like to keep my score as is.

---

### Author Response · Authors · 2024-11-21
**Thank you all for taking the time and effort to read and providing comments on our work**

We sincerely thank all reviewers for their time, effort, and valuable feedback on our work. We are particularly grateful to reviewers 5cnY and mtjz for recognizing the significance of detecting errors in LLM-generated code. We also appreciate reviewers eYV9, 5cnY, and W6to for emphasizing the importance of augmenting LLMs with program analysis tools, as well as reviewers 5cnY, mtjz, and W6to for acknowledging the practical value of our proposed method. We address the questions and concerns raised by individual reviewers in the responses below.

---

### Meta-Review · Area_Chair_UHE8 · 2024-12-24

**Metareview:**

This paper introduces REDO, which attempts to detect runtime errors in code modifications without execution by combining static analysis tools with LLMs.
The reviewers agree that the motivation and technical contributions of the paper are limited and the authors are encouraged to further improve the technical depth of the paper for the next submission.

**Additional Comments On Reviewer Discussion:**

The reviewers agree with the final decision.

---

### Decision · Program_Chairs · 2025-01-22

Reject